# NATURAL GRADIENT BAYESIAN SAMPLING AUTOMATICALLY EMERGES IN CANONICAL CORTICAL CIRCUITS

## ABSTRACT

Accumulating evidence suggests the canonical cortical circuit, consisting of excitatory (E) and diverse classes of inhibitory (I) interneurons, implements Bayesian posterior sampling. However, most of the identified circuits' sampling algorithms are simpler than the nonlinear circuit dynamics, suggesting complex circuits may implement more advanced algorithms. Through comprehensive theoretical analyses, we discover the canonical circuit innately implements natural gradient Bayesian sampling, which is an advanced sampling algorithm that adaptively adjusts the sampling step size based on the local geometry of stimulus posteriors measured by Fisher information. Specifically, the nonlinear circuit dynamics can implement natural gradient Langevin and Hamiltonian sampling of uni- and multi-variate stimulus posteriors, and these algorithms can be switched by interneurons. We also find that the non-equilibrium circuit dynamics when transitioning from the resting to evoked state can further accelerate natural gradient sampling, and analytically identify the neural circuit's annealing strategy. Remarkably, we identify the approximated computational strategies employed in the circuit dynamics, which even resemble the ones widely used in machine learning. Our work provides an overarching connection between canonical circuit dynamics and advanced sampling algorithms, deepening our understanding of the circuit algorithms of Bayesian sampling.

## 1 INTRODUCTION

The brain lives in a world of uncertainty and ambiguity, necessitating the inference of unobserved world states. The Bayesian inference provides a normative framework for this process, and extensive studies have suggested that neural computations across domains aligns with Bayesian principles, giving rise to the concept of the "Bayesian brain" (Knill & Pouget, 2004). These include visual processing (Yuille & Kersten, 2006), multi-sensory integration (Ernst & Banks, 2002), decision-making (Beck et al., 2008), sensorimotor learning (Körding & Wolpert, 2004), etc. Recent studies suggested the canonical cortical circuit may naturally implement sampling-based Bayesian inference to compute the posterior (Hoyer & Hyvärinen, 2003; Buesing et al., 2011; Aitchison & Lengyel, 2016; Haefner et al., 2016; Orbán et al., 2016; Echeveste et al., 2020; Zhang et al., 2023; Terada & Toyoizumi, 2024; Masset et al., 2022; Sale & Zhang, 2024), in that the large cortical response variability is consistent with the stochastic nature of sampling algorithms.

The canonical cortical circuit (Fig. 1A) – the fundamental computational building block of the cerebral cortex – consists of excitatory (E) neurons and various inhibitory interneurons (I) including neurons of parvalbumin (PV), somatostatin (SOM), and vasoactive intestinal peptide (VIP) (Adesnik et al., 2012; Fishell & Kepecs, 2020; Niell & Scanziani, 2021; Campagnola et al., 2022). Different interneuron classes have different intrinsic electrical properties and form specific connectivity patterns (Fig. 1B). The canonical circuit is highly conserved across a wide spectrum of vertebrate species and likely represents a common network architecture solution discovered by evolution over millions of years. Therefore, studying the algorithms underlying canonical circuits not only advances our understanding of neural computations, but also positions these circuits as building blocks for next-generation deep network models, with their clear algorithmic understanding enabling full interpretability.

The field has started to identify the algorithm of the canonical circuit. For example, a very recent study has identified the Bayesian sampling algorithm in reduced canonical circuit motifs (Sale & Zhang, 2024): the reduced circuit of only E and PV neurons can implement Langevin posterior

sampling in the stimulus feature manifold. And incorporating SOM into the circuit introduces oscillations that accelerate sampling by upgrading Langevin sampling into more efficient Hamiltonian sampling. Nevertheless, a significant gap remains between identified Bayesian sampling algorithms and the complex, nonlinear canonical circuit dynamics. A notable distinction is that canonical circuit dynamics is inherently non-linear and substantially more complex than the linear dynamics of Langevin and Hamiltonian samplings identified in previous circuit models and used in machine learning (ML) research. Rather than dismissing the added complexity as incidental to neural dynamics without computational purpose, we explore whether these nonlinear circuit dynamics may serve some advanced function. This raises a compelling question: Can nonlinear circuit dynamics implement more advanced and efficient sampling algorithms? If so, what are advanced circuit algorithms?

To address this question, we perform comprehensive theoretical analyses of the canonical circuit model composed of E neurons and two classes of interneurons (PV and SOM). Our analysis reveals that canonical circuit dynamics not only implements standard Langevin and Hamiltonian sampling as revealed in Sale & Zhang (2024), but innately incorporate the **natural gradient** (NG) to automatically adjust the step size (or the "temperature") in the circuit's **Langevin** and **Hamiltonian** sampling based on the local geometry of the posterior distribution measured by the Fisher information (FI).

Specifically, we find the total activity of E neurons monotonically increases with posteriors' FI, and dynamically control the effective sampling step size in the low-dimensional stimulus feature manifold. Remarkably, the NG sampling in canonical circuit dynamics exhibits computational strategies analogous to established numerical techniques in ML (Hwang, 2024; Girolami & Calderhead, 2011; Marceau-Caron & Ollivier, 2017). These include, **1**) the recurrent E input acts as a regularization analogous to adding a small number during FI inversion to prevent numerical instabilities (Eq. 5, $\alpha$) – a common practice in NG sampling algorithms; **2**) When coupling multiple canonical circuits interact to sample multivariate stimulus posteriors, the coupled circuit approximates the full FI matrix with its diagonal elements (Sec. 5), similar to the diagonal approximation used in scalable NG samplings. In addition, our analysis reveals that when the circuit transitions from resting state (no feedforward input) to evoked state (with feedforward input), the non-equilibrium circuit dynamics further accelerates sampling beyond the efficiency of standard NG sampling. We analytically identify the **neural annealing** strategy within canonical circuit dynamics (Fig. 2J, Eq. 3b). We also show that the canonical circuit with clear algorithm understanding can serve as a latent space sampler in deep generative models, moving a step toward biologically plausible and interpretable deep networks.

**Significance**. The present study provides the first demonstration that canonical cortical circuits with diverse classes of interneurons naturally implement natural gradient Langevin and Hamiltonian sampling. We establish a precise mapping between circuit components and computational elements of advanced sampling algorithms, bridging computational neuroscience and ML. And the canonical cortical circuit may inspire the new building block for more efficient, interpretable deep networks.

## 2  BACKGROUND: THE CANONICAL CORTICAL CIRCUIT MODEL

We consider a nonlinear canonical circuit model consisting of E neurons and two classes of interneurons (PV and SOM) (Fig. 1A), whose dynamics is adopted from a recent circuit modeling study (Sale & Zhang, 2024). This model is *biologically plausible* by reproducing tuning curves of different types of neurons (Fig. A1A-C), and is *analytically tractable* so we can directly identifying the nonlinear circuit's algorithm. Briefly, each of the $N_E$ E neurons is tuned to a preferred 1D stimulus $z = \theta_j$. The full set of these preferences, $\{\theta_j\}_{j=1}^{N_E}$, uniformly covers the whole stimulus space. E neurons are recurrently connected with a Gaussian kernel in the stimulus space (Eq. 1d). Both PV and SOM neurons are driven by E neurons but with different interactions: PV neurons deliver global, divisive normalization to E neurons (Eq. 1b), whereas SOM neurons provide local, subtractive inhibition (Eq. 1c). The whole circuit dynamics is (Sec. C for detailed explanation and construction rationale).

E: $\quad\quad \tau \dot{\mathbf{u}}_E(\theta, t) = -\mathbf{u}_E(\theta, t) + \rho \sum_X (\mathbf{W}_{EX} * \mathbf{r}_X)(\theta, t) + \sqrt{\tau \mathsf{F}[\mathbf{u}_E(\theta, t)]_+}\, \xi(\theta, t),$ (1a)

Div. norm. : $\quad \mathbf{r}_E(\theta, t) = [\mathbf{u}_E(\theta, t)]_+^2 / (1 + \rho w_{EP} r_P); \quad$ PV: $r_P = \int [\mathbf{u}_E(\theta', t)]_+^2 d\theta',$ (1b)

SOM: $\quad\quad \tau \dot{\mathbf{u}}_S(\theta, t) = -\mathbf{u}_S(\theta, t) + \rho (\mathbf{W}_{SE} * \mathbf{r}_E)(\theta, t); \quad \mathbf{r}_S(\theta, t) = g_S \cdot [\mathbf{u}_S(\theta, t)]_+,$ (1c)

Rec. weight: $\mathbf{W}_{YX}(\theta - \theta') = w_{YX} (\sqrt{2\pi} a_{XY})^{-1} \exp(-(\theta - \theta')^2 / 2a_{XY}^2),$ (1d)

Feedfwd.: $\quad \mathbf{r}_F(\theta, t) \sim \text{Poisson}[\lambda_F(\theta|z_t)], \quad \lambda_F(\theta|z_t) = R_F \exp[-(\theta - z_t)^2 / 2a^2].$ (1e)

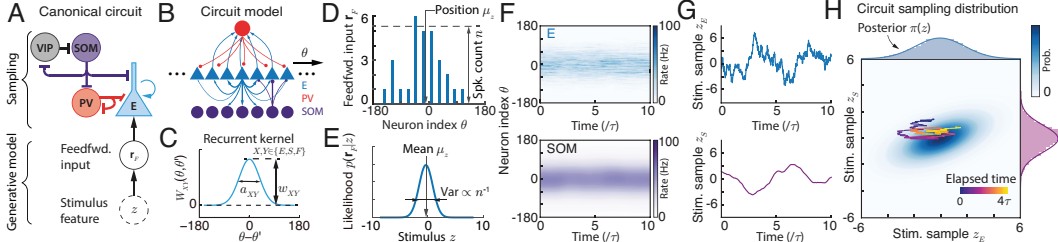

Figure 1: (A) The canonical circuit of E and diverse types of interneurons and sampling-based Bayesian inference. (B) The circuit model in the present study consists of E and two types of interneurons (PV and SOM). (C) The recurrent connection kernel between E neurons. (D) Feedforward input represented as a continuous approximation of Poisson spike trains with Gaussian tuning across the stimulus feature. (E) The feedforward input parametrically encodes the stimulus feature likelihood. (F) The population responses of E (top) and SOM neurons (bottom). (G) Stimulus feature values sampled by the E and SOM neurons respectively. (H) The network's sampling distribution read out from E and SOM neurons. The E neuron's position is regarded as stimulus feature sample $z_E$, while the sample of SOM neurons $z_S$ contributes to the auxiliary momentum variable in Hamiltonian sampling. The distribution of $z_E$ (top marginal) will be used to approximate the posterior.

$\mathbf{u}_X$ and $\mathbf{r}_X$ represent the synaptic inputs and firing rates of neurons of type $X$ respectively. In Eq. (1a), the neuronal types $X \in \{E, F, S\}$ representing inputs from E neurons, sensory feedforward inputs (Eq. 1e), and SOM neurons (Eq. 1c) respectively. $[x]_+ = \max(x, 0)$ is the negative rectification. E neurons receive internal Poisson variability with Fano factor $\mathsf{F}$, mimicking stochastic spike generation that can provide appropriate internal variability for circuit sampling (Zhang et al., 2023). In particular, $g_S$ is the "gain" of SOM neurons and can be modulated (see Discussion), which is the key circuit mechanism to flexibly switch between static inference and dynamic inference with various speeds.

To facilitate math analysis, the above dynamics consider infinite number of neurons in theory ($N_E \to \infty$), then the summation of inputs from other neurons $\theta_j$ becomes an integration (convolution) over $\theta$, e.g., $(\mathbf{W} * \mathbf{r})(\theta) = \int \mathbf{W}(\theta - \theta')\mathbf{r}(\theta')d\theta'$, while our simulations take finite number of neurons. $\rho = N_E/2\pi$ is the neuronal density in the stimulus feature space, a factor in discretizing the integral.

### 2.1 THEORETICAL ANALYSIS OF THE CANONICAL CIRCUIT DYNAMICS

It has established theoretical approach to obtain **analytical** solutions of the nonlinear recurrent circuit dynamics considered in the present study (Fung et al., 2010; Wu et al., 2016; Zhang & Wu, 2012; Sale & Zhang, 2024), including attractor states, full eigenspectrum of the perturbation dynamics, and the projected dynamics onto the dominant eigenmodes. These analytical solutions are essential to identify the circuit's Bayesian algorithms. Below, we briefly introduce the key steps and results of the theoretical analysis, with detailed math calculations in Sec. C.

**Attractors.** E neurons in canonical circuit dynamics have the following attractor states with a bump profile over the stimulus feature space (Fig. A1; Sec. C),

$$\bar{\mathbf{u}}_E(\theta) = \bar{U}_E \exp[-(\theta - \bar{z}_E)^2/4a^2], \quad \bar{\mathbf{r}}_E(\theta) = \bar{R}_E \exp[-(\theta - \bar{z}_E)^2/2a^2]. \tag{2}$$

Similar bump attractor states exist for SOM neurons (Eq. E2 ). In contrast, PV neurons don't have a spatial bump profile since their interactions with E neurons are unstructured (Eq. 1b).

**Dimensionality reduction for stimulus sampling dynamics.** The perturbation analysis reveals that the first two dominant eigenmodes of the circuit dynamics correspond to the change of bump position $z_E$ and the bump height $U_E$ respectively (Sec. C, (Fung et al., 2010; Wu et al., 2016)), and similarly for SOM neurons. We project the E dynamics (Eq. 1a) onto the above two dominant eigenvectors (calculating the inner product of the circuit dynamics and the eigenvectors), yielding the governing dynamics of the $z_E$ and $U_E$ (the projection of SOM neurons will be shown later in Sec. 6 and E),

$$\text{Position}: \quad \dot{z}_E \approx (\tau U_E)^{-1} U_{EF}(\mu_z - z_E) + \sigma_z(\tau U_E)^{-1/2}\xi_t, \quad (U_{XY} = \rho w_{XY} R_Y/\sqrt{2}) \tag{3a}$$

$$\text{Height}: \quad \dot{U}_E \approx \tau^{-1}[-U_E + U_{EE} + U_{EF}] + \sigma_U(\tau^{-1}U_E)^{1/2}\xi_t, \tag{3b}$$

where $U_{XY}$ is the population input height from population $Y$ to $X$. $\sigma_z^2 = 8a\mathsf{F}/(3\sqrt{3\pi})$ and $\sigma_U^2 = \mathsf{F}/(\sqrt{3\pi}a)$ are constants that don't change with network activities. The approximation comes

from omitting negligible nonlinear terms in the circuit dynamics (Sec. C.3). Our following theoretical analysis on circuit algorithms will be based on the above two equations.

## 2.2 BACKGROUND: NATURAL GRADIENT BAYESIAN SAMPLING

Amari's natural gradient is a well-known method to adaptively adjust the sampling step size based on the local geometry characterized by the Fisher information (FI) $G(z)$ (Amari, 1998; Amari & Douglas, 1998; Amari, 2016; Girolami & Calderhead, 2011) (see details in Sec. (B.2)),

$$G(z) = -\mathbb{E}_{p(\mathbf{r}_F|z)}[\nabla_z^2 \ln \pi(z)] \tag{4}$$

For a Gaussian distribution $\mathcal{N}(z|\mu, \Lambda^{-1})$, the FI will be its precision $\Lambda$ and doesn't depend on $z$.

**Natural gradient Langevin sampling (NGLS).** The FI is used to determine the step size of the Langevin sampling dynamics to sample the posterior $\pi(z)$ (Girolami & Calderhead, 2011),

$$\dot{z} = \tau_L^{-1} \nabla \ln \pi(z) + (2\tau_L^{-1})^{1/2}\xi_t, \quad \text{where } \tau_L = \eta[G(z) + \alpha]. \tag{5}$$

$\alpha$ is a small positive constant acting as a regularization term to improve numerical stability in inverting the FI (Hwang, 2024; Marceau-Caron & Ollivier, 2017; Wu et al., 2024), which is widely used in ML. $\eta$ is a small constant similar to the inverse of "learning rate". In the naive Langevin sampling, $\tau_L$ is fixed rather than proportional to the FI. In the NG Langevin sampling, the $\tau_L$ scales with the FI. If the distribution is widely spread out, the sampling step size will be larger, allowing for faster exploration of the space. Conversely, if the distribution is sharply peaked, the sampling step size will be smaller to explore the local region more thoroughly.

**Natural gradient Hamiltonian sampling (NGHS).** It defines a Hamiltonian function $H(z, p)$ where the momentum distribution $\pi(p|z)$'s variance (rather than precision) is proportional to the FI $G(z)$.

$$H(z, p) = -\ln \pi(z) - \ln \pi(p|z), \quad \pi(p|z) = \mathcal{N}[p|0, G(z)]. \tag{6}$$

The NGHS dynamics with friction is governed by (Girolami & Calderhead, 2011; Ma et al., 2015),

$$\frac{d}{dt}\begin{bmatrix} z \\ p \end{bmatrix} = -\begin{bmatrix} 0 & -\tau_H^{-1} \\ \tau_H^{-1} & \gamma \end{bmatrix}\begin{bmatrix} \nabla_z H \\ \nabla_p H \end{bmatrix} + \sqrt{2}\begin{bmatrix} 0 \\ \gamma^{1/2} \end{bmatrix}\boldsymbol{\xi}_t \tag{7}$$

where $\tau_H$ is the time constant of the Hamiltonian sampling, and $\gamma$ is the friction that dampens momentum. The Hamiltonian dynamics with friction can be interpreted as a Langevin dynamics added into the momentum dynamics (Chen et al., 2014; Ma et al., 2015). When $\gamma = 0$, Eq. (7) reduces into the naive Hamiltonian dynamics. Our following analysis will show the canonical circuit can automatically implement the natural gradient Langevin sampling and Hamiltonian sampling.

## 3 FROM CIRCUIT DYNAMICS TO BAYESIAN SAMPLING

In our framework, the stage from external stimulus $z$ to the feedforward input $\mathbf{r}_F$ is regarded as a generative process (Fig. 1A), and then the circuit dynamics (Eqs. 1a and 1c) effectively performs Bayesian sampling dynamics to compute the stimulus posterior, $\pi(z) \equiv p(z|\mathbf{r}_F) \propto p(\mathbf{r}_F|z)p(z)$. We hypothesize that the circuit computes **subjective** posterior distributions $\pi(z)$ based on its internal generative model (Lange et al., 2023), implicitly assuming the subjective prior in brain's neural circuits matches the *objective* prior (usually not known precisely) of the world. With this hypothesis, we treat the canonical circuit, strongly supported by experiments, as a "ground truth", and aim to identify the circuit's internal generative model and its Bayesian sampling algorithms.

**Subjective prior** $p(z)$. We will leave the subjective prior $p(z)$ unspecific for now and will find it through the analysis of the circuit dynamics. This will be shown later in the Eqs. (10 and 13).

**Stimulus likelihood** $L(z)$. The stochastic feedforward input from the stimulus $z$ (Eq. 1e) naturally specifies the stimulus likelihood that is calculated as a Gaussian likelihood (see Sec. C.4),

$$\mathcal{L}(z) \propto p(\mathbf{r}_F|z) = \prod_\theta \text{Poisson}[\lambda_F(\theta|z)] \propto \mathcal{N}(z|\mu_z, \Lambda^{-1}),$$

$$\text{where } \mu_z = \sum_j \mathbf{r}_F(\theta_j)\theta_j / \sum_j \mathbf{r}_F(\theta_j), \quad \Lambda = a^{-2}\sum_j \mathbf{r}_F(\theta_j) = \sqrt{2\pi}\rho a^{-1}R_F. \tag{8}$$

The mean $\mu_z$ and precision $\Lambda$ are geometrically regarded as $\mathbf{r}_F$'s location and height respectively (Fig. 1D-E). A single snapshot of $\mathbf{r}_F$ *parametrically* conveys the stimulus likelihood $p(\mathbf{r}_F|z)$, in

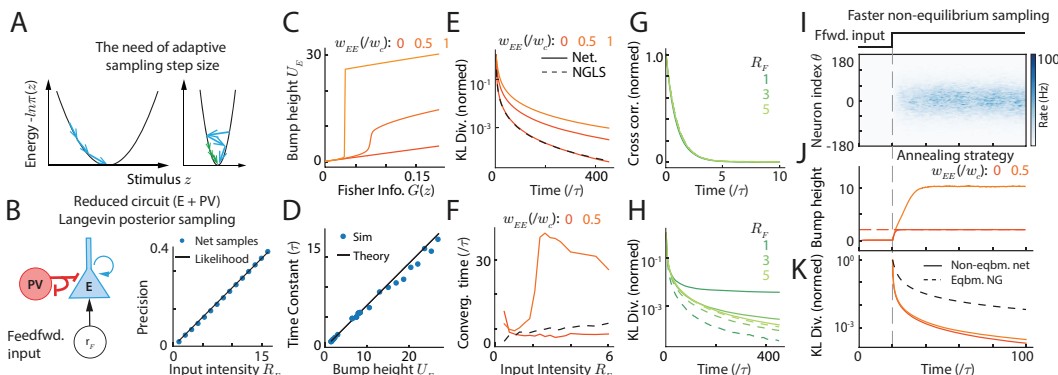

Figure 2: NG Langevin sampling in the reduced circuit (E and PV). (A) The need for adaptive step size to sample different posteriors. (B) The reduced circuit with fixed weights flexibly samples posteriors with different uncertainties. (C-D) E bump height $U_E$ increases with posterior FI (C) and determines the sampling time constant in the stimulus feature manifold. (E-F) The sampling convergence with recurrent weight $w_{EE}$ that acts as a regularizer (Eq. 5). (G-H) The sampling convergence at different posterior uncertainties. (I-K) Non-equilibrium sampling further accelerates convergence. The non-equilibrium population responses (I), bump height (J), and the KL divergence (K) from the resting state (no feedforward input) to evoked state.

that all likelihood parameters are read out from $\mathbf{r}_F$. In particular, the Gaussian stimulus likelihood resulted from the Gaussian profile of feedforward input tuning $\lambda_F(\theta|z)$ (Eq. 1e, Ma et al. (2006).

**Circuit's stimulus posterior FI** comes from the likelihood (Eq. 8) and the prior (unspecified now),

$$G(z) = \Lambda + \nabla_z^2 \ln p(z) = \sqrt{2\pi}\rho a^{-1} R_F + \nabla_z^2 \ln p(z). \tag{9}$$

Our following analysis will focus on connecting the circuit dynamics on the position and height subspace (Eqs. 3a and 3b) to the NG sampling dynamics (Sec. 2.2), to identify how the circuit implements NG Langevin and Hamiltonian sampling. To facilitate understanding, we will start from the reduced circuit model without SOM neurons (Sec. 4 - 5) and then add the SOM back (Sec. 6).

## 4 A REDUCED CIRCUIT WITH E AND PV NEURONS: NG LANGEVIN SAMPLING

To facilitate understanding, we first present how the circuit realizes the naive Langevin sampling (Sale & Zhang, 2024), then conduct further analyses to reveal its mechanism of NG Langevin samplings.

### 4.1 NAIVE LANGEVIN SAMPLING IN THE REDUCED CIRCUIT

To analyze the circuit Langevin sampling, we convert the bump position $z_E$ dynamics (Eq. 3a) into a Langevin sampling form by expressing its drift term as the log-likelihood gradient,

$$\dot{z}_E = (\tau U_E)^{-1}\lambda_z \nabla \ln \mathcal{L}(z_E) + \sigma_z(\tau U_E)^{-1/2}\xi_t, \text{ with } \nabla \ln \mathcal{L}(z) = \Lambda(\mu_z - z), \lambda_z = \frac{w_{EF}a}{2\sqrt{\pi}}, \tag{10}$$

where the feedforward input strength $U_{EF}$ (Eq. 3a) is proportional to the likelihood precision $\Lambda$, i.e., $U_{EF} \propto w_{EF}R_F \propto w_{EF}\Lambda$ (Eq. 8). Notably, the drift and diffusion terms in Eq. (10) share the same factor $\tau U_E$, a necessary condition for Langevin sampling (Eq. 5). Then we investigate how the circuit realizes Langevin sampling by comparing Eqs. (10 and 5), and study its sampling structure.

**Uniform (uninformative) circuit prior**. It is uniform because the drift term in Eq. (10) is the stimulus likelihood $\mathcal{L}(z)$ gradient, due to the translation-invariant recurrent weights (Eq. 1d). This result is consistent with the previous study (Zhang et al. (2023); Sale & Zhang (2024); see Discussion).

**The circuit sampling only constrains feedforward weight** $w_{EF}$. It requires the ratio $\sigma_z^2/\lambda_z = 2$ in Eq. (10) which only constrains the feedforward weight as $w_{EF}^* = \sqrt{\pi}\sigma_z^2/a = (2/\sqrt{3})^3\mathsf{F}$, irrelevant with other circuit weights like $w_{EE}$ and $w_{EP}$ as long as the circuit dynamics is stable. This suggests the *robust* circuit sampling and *no fine-tuning* of circuit parameters is needed.

**Flexible sampling the whole likelihood family**. Once the feedforward weight is set at $w_{EF}^*$, the circuit with fixed weights flexibly samples likelihoods with different means and uncertainties, because in Eq. (10) the $\lambda_z$ and $\sigma_z$ are invariant with circuit activities, and $\tau U_E$ is a free parameter without changing the equilibrium sampling distribution. And then the bump position $z_E$ dynamics will automatically sample the corresponding likelihood that is parametrically represented by the *instantaneous* feedforward input $\mathbf{r}_F$ (Eq. 8). This is also confirmed by our simulation (Fig. 2B).

### 4.2 Natural gradient Langevin sampling in the reduced circuit

The NG Langevin sampling requires the sampling time constant increases with the FI $G(z)$ (Eq. 5). Meanwhile, the time constant of the circuit's bump position $z_E$ dynamics is proportional to the bump height $U_E$ (Eq. 3b and 10) and is confirmed by circuit simulation (Fig. 2D). Thus we analyze the relation between $U_E$ and the FI. For simplicity, we first focus on the equilibrium mean of $U_E$ (averaging Eq. 3b), and the identified NGLS parameters in the circuit are shown in Fig. 4E.

$$\bar{U}_E = U_{EE} + U_{EF}, \quad U_{EF} = \rho w_{EF} R_F / \sqrt{2} = \lambda_z \cdot G(z) = \lambda_z \Lambda. \tag{11}$$

**E bump height $U_E$ encodes Fisher information.** The feedforward input height $U_{EF}$ is proportional to the likelihood FI $G(z)$ (Eq. 9, uniform prior), making the mean bump height $\bar{U}_E$ increase with $G(z)$. This is also confirmed by the circuit simulation (Fig. 2C). Consequently, the bump height $\bar{U}_E$ effectively represents the stimulus FI and in turn scales the time constant of the circuit sampling $z_E$ dynamics (Eq. 10, Fig. 2D), enabling the NGLS in the circuit.

**The recurrent E input (weight) acts as a regularizer**. Comparing Eqs. (11 and 5), the recurrent input strength $U_{EE}$ acts as a role of the regularization coefficient $\alpha$, improving the numerical stability in inverting the FI when it is small or ill-conditioned (Hwang, 2024; Marceau-Caron & Ollivier, 2017; Wu et al., 2024). Without recurrent E weight ($U_{EE} = 0$ via setting $w_{EE} = 0$), the circuit sampling behaves similarly with the NGLS (Fig. 2E). Including recurrent weights enlarges the sampling time constant, slowing down the sampling as suggested by our theory (Fig. 2E). Nevertheless, with extremely small FI, the circuits with higher recurrent weights have faster convergence (Fig. 2F, leftmost part), because the recurrent E input stabilizes the inversion of very small FI. Moreover, NGLS is characterized by the invariant temporal correlation of samples with the posterior uncertainties (controlled by input intensity $R_F$), which is also confirmed in the circuit simulation (Fig. 2G).

**The flexible scaling with various posterior FI.** The canonical circuit model with fixed weights flexibly scales its sampling time constant (determined by $\bar{U}_E$, Eq. 3a) with various posteriors FI (controlled by the feedforward input rate $R_F$), all of which is *automatically* completed by the recurrent dynamics without the need of changing circuit parameters. For example, increasing $R_F$ increases the bump height $U_E$ (Eq. 3b) that leads to a larger sampling time constant, and meanwhile it changes the equilibrium sampling distribution of the circuit (Eqs. 8 and 3a).

### 4.3 Non-equilibrium circuit dynamics accelerates natural gradient sampling.

Our analysis so far concentrates on the equilibrium mean $\bar{U}_E$ (Eq. 11). We now extend to the non-equilibrium dynamics (Eq. 3b), particularly during the transient response immediately following the onset of a stimulus. After receiving a $\mathbf{r}_F$, the $U_E$ will gradually grow up until the equilibirum state (Fig. 2I-J). And meanwhile, the sampling step size will gradually decreases in that a larger $U_E$ leads to larger sampling constant and therefore smaller step size. This is similar to the **annealing** in stochastic computation. The larger sampling step size during the non-equilibrium implies the circuit can sample faster than the equilibrium state, confirmed by simulation (Fig. 2K). Furthermore, $U_E$ temporal trajectory (Fig. 2J) describes circuit's **annealing strategy**, governed by Eq. (3b). This intrinsic annealing schedule is an emergent property of the circuit's nonlinear dynamics.

## 5 Coupled circuits: NGLS of multivariate posteriors

The brain needs to sample multivariate stimulus posteriors, which can be implemented by a **decentralized** system consisting of multiple coupled canonical circuit modules (Fig. 3A, Zhang et al. (2016; 2023); Raju & Pitkow (2016)). Each circuit module $m$ receives a feedforward input stochastically evoked by a 1D stimulus $z_m$ (Fig. 3), and the cross-talk between circuits enables them to read out the circuit prior, and eventually each circuit $m$ samples the corresponding stimulus $z_m$ distributedly. As

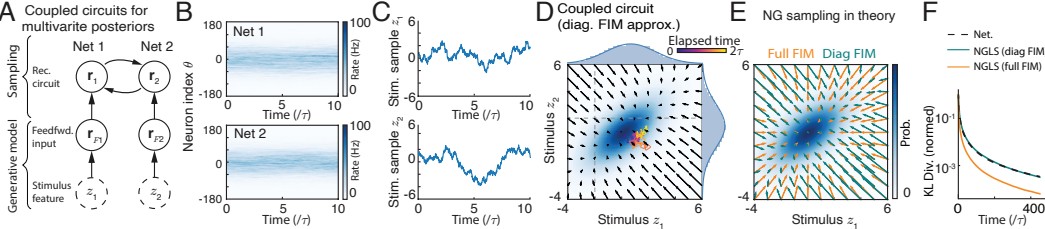

Figure 3: Coupled canonical circuits sample multivariate posteriors, with each circuit $m$ sampling the corresponding marginal posterior of $z_m$. (A) The structure of decentralized circuit with each consisting of E and PV neurons (the same as Fig. 2B). (B-C) The spatiotemporal E neuronal responses in two circuits (B) and the decoded stimulus samples (C). (D) When concatenating the samples from two circuits together, we recover the bivariate sampling distribution. Vector field: the drift term of the sampling dynamics in the circuit. (E) The vector field of natural gradient sampling with full FIM and diagonal FIM approximation. (F) The convergence speed in the decentralized circuit. The diagonal FIM approximation is scalable in high dimensions, while paying the cost of slower sampling speed.

a proof of concept, we consider the *smallest* decentralized system of two coupled circuits to sample bivariate posteriors (Fig. 3A). The sampling of higher dim. posteriors can be extended by inserting more circuit modules, with the number of circuit modules determined by the stimulus dimension.

We investigate how the coupled circuits implement bivariate posteriors' NGLS, and what kind of approximation, if there is any, is used in the circuit. The theoretical analysis of the two coupled circuits is similar to a single circuit module, but we perform the analysis on each circuit module individually, yielding the new position and height dynamics (details at Sec. D),

$$\text{Position}: \quad \dot{\mathbf{z}}_E = (\tau \mathbf{D_U})^{-1}\big[ -\mathbf{Lz}_E + \mathbf{U}_{EF} \circ (\boldsymbol{\mu_z} - \mathbf{z}_E)\big] + \sigma_z(\tau \mathbf{D_U})^{-1/2}\boldsymbol{\xi}_t. \quad (12a)$$

$$\text{Height}: \quad \dot{\mathbf{U}}_E = \tau^{-1}\big( -\mathbf{U}_E + \mathbf{U}_{EE} + \mathbf{U}_{EF}\big) + \sigma_U(\tau^{-1}\mathbf{D_U})^{1/2}\boldsymbol{\xi}_t. \quad (12b)$$

$\mathbf{z}_E = (z_1, z_2)^\top$ is two circuits' E bump positions. Similarly for $\boldsymbol{\mu_z}$ and $\mathbf{R}_F$ (feedfwd. input position and intensity respectively), and $\mathbf{U}_E$ and $\mathbf{R}_E$ (bump height of synaptic input and firing rate respectively). $\mathbf{D}_U = \text{diag}(\mathbf{U}_E)$ is a diagonal matrix of $\mathbf{U}_E$. The $\circ$ denotes the element-wise product.

**The associative bivariate stimulus prior.** The $\mathbf{z}_E$ dynamics (Eq. 12a) has a new term, i.e., $-\mathbf{Lz}_E$, which can be linked to the internal stimulus prior (omitting the subscript $EE$ of $U$ for clarity below).

$$\nabla \ln p(\mathbf{z}) = -\mathbf{Lz}_E, \quad \mathbf{L} = \begin{pmatrix} U_{12} & -U_{12} \\ -U_{21} & U_{21} \end{pmatrix} \quad \Leftrightarrow \quad p(\mathbf{z}) \propto \exp\left(-\mathbf{z}_E^\top \mathbf{Lz}_E/2\right). \quad (13)$$

Hence the coupling matrix $\mathbf{L}$ is the prior's precision matrix (see Sec. D.2). For ease of understanding, we expand the bivariate prior as $p(z_1, z_2) \propto \exp\left[-\Lambda_{12}(z_1 - z_2)^2/2\right]$, with $\Lambda_{12} \propto (U_{12} + U_{12})/2$. The coupled circuit stores an associative (correlational) stimulus prior with each marginal uniform, consistent with previous studies (Sale & Zhang, 2024; Zhang et al., 2023). The identified correlational prior is confirmed by numerical simulation, where the actual sampling distribution of the circuit dynamics matches the subjective posterior predicted via the identified prior (Fig. A2).

**Diagonal Fisher information approximation**. Given the identified circuit's prior, we calculate the Fisher information matrix (FIM) $\mathbf{G}(\mathbf{z})$ and compare it with the actual sampling time constants,

$$\text{FI:} \quad \mathbf{G}(\mathbf{z}) = \lambda_z^{-1}\begin{pmatrix} U_1 & -U_{12} \\ -U_{21} & U_2 \end{pmatrix} \quad \text{vs.} \quad \text{Circuit:} \quad \mathbf{D}_U = \begin{pmatrix} U_1 & \\ & U_2 \end{pmatrix}. \quad (14)$$

The time constant matrix $\mathbf{D}_U$ in the circuit dynamics is proportional to the diagonal elements of the FIM, i.e., $\mathbf{D_U} \propto \text{diag}[\mathbf{G}(z)]$, suggesting the circuits utilize the **diagonal approximation** of the FIM to scale the Langevin sampling step size. The diagonal FIM approximation is widely used in ML, as a trade-off of computational efficiency and accuracy (Amari, 1998; Amari & Douglas, 1998; Amari, 2016; Wu et al., 2024), where the full FIM is hard to estimate.

To illustrate the effect of diagonal FIM approximation on sampling dynamics, we plot the vector field of NGLS with full FIM and diagonal FIM respectively (Fig. 3D). All vector fields under full FIM directly point to the posterior mean, while the ones under diagonal FIM are curled along the long axis

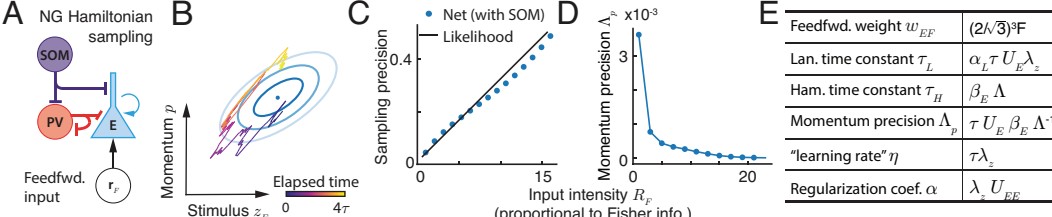

Figure 4: Natural gradient Hamiltonian sampling in the augmented circuit with E, PV, and SOM neurons. (A) The circuit structure. (B) The decoded trajectory of stimulus sample $z_E$ and momentum $p$ exhibits an oscillation pattern, which is a characteristic of Hamiltonian sampling. The momentum $p$ is a weighted average of sample $z_E$ and $z_S$ as shown in Fig. 1H. (C) The circuit with fixed weights can sample posteriors with different uncertainties. (D) The momentum precision decreases with the Fisher information controlled by feedforward input strength, satisfying the requirement of natural gradient Hamiltonian sampling. (E) A table summarizing the circuit' sampling parameters.

of the posterior. Meanwhile, the curled vector fields also exist in coupled circuits' sampling dynamics, confirming diagonal FIM approximation in the circuit (Fig. 3C). The diagonal FIM simplifies the computation, while paying the cost of sampling speed (Fig. 3E and Fig.A4).

## 6 THE CIRCUIT WITH SOM NEURONS: NG HAMILTONIAN SAMPLING

We investigate the Bayeisan sampling in the augmented circuit model with SOM neurons providing structured inhibition to E neurons (Eq. 1a). The SOM's structured inhibition can add the Hamiltonian sampling component in the circuit (Sale & Zhang, 2024). We further analyze whether the augmented circuit with SOM neurons implements the natural gradient Hamiltonian sampling (NGHS). For simplicity, we consider a augmented circuit model to sample a univariate stimulus posterior. Similarly, we derive the eigenvectors of the SOM's dynamics and then project the dynamics on dominant eigenvectors (see details in Sec. E). The position dynamics of the E and SOM neurons are,

$$\text{E:} \qquad \tau_E \dot{z}_E = \underbrace{[U_{ES}(z_S - z_E) + \alpha_H U_{EF}(\mu_z - z_E)]}_{\text{Momentum } p, \text{ (Hamiltonian part)}} + \underbrace{[\alpha_L U_{EF}(\mu_z - z_E) + \sigma_z \sqrt{\tau_E}\xi_t]}_{\text{Langevin part}}, \quad (15a)$$

$$\text{SOM:} \ \tau_S \dot{z}_S \approx U_{SE}(z_E - z_S), \qquad (\tau_X = \tau U_X) \tag{15b}$$

To understand the circuit's sampling dynamics, the $z_E$ dynamics (Eq. 15a) is decomposed into the drift terms from Langevin and Hamiltonian parts with $\alpha_H + \alpha_L = 1$, and the momentum $p$ is defined as a mixture of $z_E$ and $z_S$. Transforming the $(z_E, z_S)$ dynamics (Eqs. 15a - 15b, Fig. 1H) into the $(z_E, p)$ dynamics (Fig. 4B) shows a mixture of Langevin and Hamiltonian sampling in the circuit,

$$\frac{d}{dt}\begin{bmatrix} z_E \\ p \end{bmatrix} = -\begin{bmatrix} \alpha_L \lambda_z \tau_E^{-1} & -\beta_E \Lambda^{-1} \\ \beta_E \Lambda^{-1} & \tau U_E \beta_p \beta_E \Lambda^{-1} \end{bmatrix} \begin{bmatrix} -\nabla_z \ln \pi(z_E) \\ (\tau_E \beta_E)^{-1}\Lambda \cdot p \end{bmatrix} + \begin{bmatrix} \sigma_z \tau_E^{-1/2} \\ \sigma_p \end{bmatrix} \boldsymbol{\xi}_t \tag{16}$$

where $\beta_p$, $\beta_E$ and $\sigma_p$ are functions of the coefficients in Eq. (15a) (details at Eq. E16). And the momentum $p$ dynamics has a friction term (Eq. 16), corresponding to a Langevin component.

**A line manifold in weight space for Hamiltonian sampling.** It requires the ratio between the drift and diffusion coefficients are the same as the Langevin (Eq. 5) and Hamiltonian sampling (Eq. 7). Specifically, it requires 1) $\alpha_L \lambda_z \tau_E^{-1} = \sigma_z^2 \tau_E^{-1}/2$ and 2) $\tau_E \beta_p \beta_E \Lambda^{-1} = \sigma_p^2/2$. Solving the two constraints, we can derive the requirement of circuit weights for Hamiltonian sampling,

$$\left(U_E^{-1}R_S\right) \cdot w_{ES} - \left[(1-\alpha_L)U_E^{-1}R_F\right] \cdot w_{EF} = \left[Q(\alpha_L)U_S^{-1}R_E\right] \cdot w_{SE}. \tag{17}$$

$Q(\alpha_L)$ is nonlinear with $\alpha_L$ and is invariant with network activities (Eq. E21). $U_X$ and $R_X$ are the height of the population synaptic input and firing rate of neurons $X$ (Eq. 2). Eq. (17) suggests a **line manifold** in the circuit's weight space $(w_{ES}, w_{EF})$ for correct posterior sampling, confirmed by numerical simulation (Fig. A3). Moreover, once circuit weights are set within the line manifold, the circuit with fixed weights can flexibly sample posteriors with various uncertainties (Fig. 4C).

**Natural gradient Hamiltonian sampling.** Implementing NGHS requires the precision of the momentum $p$ to be inversely proportional to posterior's FI, $G(z)$ (Eq. 7). To verify this, we calculate

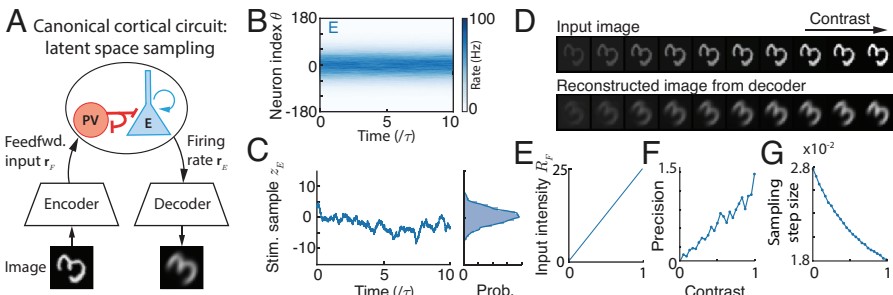

Figure 5: Canonical circuit sampling in an autoencoder's latent space. (A) The network structure. The encoder and decoder may refer to the feedforward and feedback neural pathways in the cortical hierarchy. Each stimulus feature sample can be fed into the decoder to generate the image reconstruction. (B-C) The E population responses (B) and the decoded stimulus samples (C). (D) An example image from MNIST dataset with different contrast, and the reconstructed images. (E-G) Increasing the contrast increases the feedforward input intensity (E), sampling precision in the canonical circuit (F) and meanwhile decreases the sampling step size (G).

the momentum distribution $\pi(p|z)$ in the circuit (comparing Eqs. 16 and 7),

$$-\nabla \ln \pi(p|z) = (\tau_E \beta_E)^{-1} \Lambda \cdot p \;\Rightarrow\; \pi(p|z) = \mathcal{N}(p|0, \Lambda_p^{-1}), \text{ where } \Lambda_p = (\tau_E \beta_E)^{-1} \Lambda \quad (18)$$

We analyze the momentum precision $\Lambda_p$ in the circuit dynamics. Since $\beta_E$ is a complex, quadratic function of neuronal responses, we then use *order* analysis to provide insight (details at Sec. E.4). In the circuit dynamics, we calculate $\beta_E \sim \mathcal{O}(G(z))$, $U_E \sim \mathcal{O}(G(z))$, and $G(z) = \Lambda$, and then we have $\Lambda_p \propto \mathcal{O}(G(z)^{-1})$, suggesting the momentum precision $\Lambda_p$ decreases with posterior's FI and satisfies the requirement of NGHS. This is confirmed by simulation results (Fig. 4D).

## 7 CANONICAL CIRCUIT: NG LATENT SPACE SAMPLER IN AUTOENCODERS

The exact, analytical mapping between nonlinear canonical circuit dynamics and the sampling algorithm not only enhances our algorithmic understanding of circuit computation, but also the canonical circuit model can be a novel, biologically plausible building block for machine learning models. As a proof-of-concept example, we embed the Gaussian posterior sampling canonical circuits (Fig. 2B) into the latent space of deep generative models like variational autoencoders (VAE) (Kingma & Welling, 2013) (Fig. 5A). In the autoencoder framework, its encoder and decoder can be regarded as the feedforward and feedback neural pathways in the cortical hierarchy, which link the data with complex distributions into simple Gaussian distributions in the latent space.

Since the latent space sampling in VAE is directly given without training, we can directly connect our handcrafted canonical circuit (Fig. 2B) with an encoder and a decoder, avoiding the time-consuming training of the recurrent circuit. Given the canonical circuit and its parametric form of feedforward input (Eqs. 1a - 1e), we directly train an encoder in a supervised way that transforms MNIST hand-written digit images into the two parameters of feedforward neural inputs, i.e., input strength $R_F$ (Eq. 1e) and location $\mu_z$ (Eq. 8) which represent the physical attributes of image contrast and orientation respectively. This is supported by neuroscience studies that the V1 population bump responses' strength increases with image contrast, while their bump response location represents orientation (Ben-Yishai et al., 1995; Rubin et al., 2015). Then we separately train the decoder that converts E neurons' firing rate $\mathbf{r}_E(\theta)$ into reconstructed images (see Appendix Sec. F.4 for details).

The discovered natural gradient sampling in canonical circuits still functions in the autoencoder framework. Increasing the image contrast increases the feedforward input strength (Fig. 5E), which in turn increases the likelihood precision and the E neuronal firing rate in the circuits (Fig. 5D). And the model successfully reconstructs the images with different contrasts. When examining the sampling in the canonical circuit in the latent space (Fig. 5B-C), we find the precision of the samples generated in the circuit increases with image contrast (Fig. 5F), and importantly, the sampling step size decreases with the contrast (Fig. 5G), suggesting the natural gradient sampling is reserved.

## 8 CONCLUSION AND DISCUSSION

The present theoretical study for the first time discovers that the canonical circuit dynamics with E and two classes of interneurons (PV and SOM) innately implement **natural gradient** sampling of stimulus posteriors, deepening our understanding of circuit computations. The circuit samples stimulus posterior in the stimulus manifold that is geometrically regarded as the E neurons' bump position. And we find the E bump height encodes the FI of the stimulus posterior, and determine the time constant of bump position's sampling dynamics. We find the **non-equilibirum** dynamics of the E bump height can further accelerate sampling, and explicitly identify the circuit **annealing** strategy (Eq. 3b). Remarkably, we discover the circuit dynamics also utilizes computational approximations widely used in ML algorithms, including the regularization coefficient for inverting FI (Eq. 11) and the diagonal FI matrix approximation in multivariate cases (Eq. 12b), which provides a direct evidence to validate the biological plausibility of artificial ML algorithms. Our work unprecedentedly links the canonical circuit with classes of interneurons to natural gradient sampling and related approximation strategies, providing deep, mechanistic insight into circuit sampling algorithms. At last, our proof-of-concept example suggests the canonical circuit can be a biologically plausible and interpretable building block for latent space sampling in deep generative models.

**Preliminary experimental support of NG sampling**. Our NG sampling circuit specifically predicts that the magnitude of the E neurons' responses (the bump height $U_E$, Eq. 3b), is inversely proportional to the step size of the trajectory in the stimulus feature subspace (bump position $z_E$, Eq. 3a). This is supported by experiments from hippocampal place cells where the step size of the decoded spatial trajectories (akin to our $z_E$) was found to be negatively correlated with population firing rate (Pfeiffer & Foster, 2015), providing a necessary condition for validating circuit NG sampling.

**Comparison with previous studies. First**, In our best knowledge, only one study investigated the NG sampling in recurrent networks (Masset et al., 2022), while it is difficult to make direct and "fair" comparison since the network models in two studies are different: The previous study considers a spiking network without explicit defining neuron types, while the present study considers a rate-based network with diverse classes of interneurons (PV and SOM) that has rich repertoire of realizing various NG sampling algorithms (Langevin and Hamiltonian). From functional perspective, our circuit with *fixed weights* can flexibly realize NG sampling for posteriors with different mean and uncertainties, whereas it remains unknown whether this flexiblility holds in the previous study. **Second**, our circuit model builds upon a recent work (Sale & Zhang, 2024) that discovered the conventional Langevin and Hamiltonian sampling in the canonical circuit. Our work takes one step further and finds the same circuit can innately realize NG Langevin and Hamiltonian sampling, which is a fundamentally deeper result after more comprehensive theoretical analysis of the circuit by additionally projecting the circuit dynamics onto the second dominant height mode (Eq. 3b).

**Limitations, generalizations, and future directions. First**, the proposed circuit model doesn't include VIP neurons (Fig. 1), which are likely act as a "knob" modulating the SOM gain ($g_S$, Eq. 1c) to adjust circuit sampling speed and the momentum (Sec. E.4). **Second**, Our canonical circuit model, widely used in neuroscience, only stores a uniform (marginal) prior for each stimulus as a result of an ideal case that neurons are uniformly tiling the stimulus manifold and translation-invariant recurrent weights (Eq. 1d). This implies the circuit has to break the neuronal homogeneity on the stimulus manifold to store a non-uniform (marginal) prior (Ganguli & Simoncelli, 2010). Regarding the circuit mechanism, the tuning heterogeneity for non-uniform marginal prior may come from 1) an external prior input that may from higher cortex (Appendix Sec.G.1), or 2) internally stored in the recurrent weights in the network model which can be realized by introducing an extra heterogeneous recurrent weight component superimposed on the translation-invariant recurrent weight matrix. **Third**, although our circuit with fixed weights automatically scale its sampling time constant with various posteriors' FI, for each posterior it uses a globally homogeneous FI because the Gaussian posteriors have homogeneous curvature. In principle, we can change the profile of the recurrent kernel, and then the circuit can sample other posteriors in the exponential family with locally dependent FI. We show a von Mises case in Sec. G.2. **Fourth**, we can introduce bump height $U_E$ oscillations with larger PV inhibitory weight $w_{EP}$, and then the circuit has the potential to implement cosine-profile annealing. **Fifth**, to implement the NG sampling of general distributions, one possibility is our circuit samples baseline Gaussian distributions, and a feedforward decoder network map the base distribution into arbitrary distributions. Preserving the NG sampling in the space of arbitrary distribution probably requires the diffeomorphism of the decoder network. All of these form our future research.

## 9 REPRODUCIBILITY STATEMENT

All analytical calculations of the nonlinear circuit dynamics are detailed from Appendix Sec. B - E. Below is a list of the Appendix sections and their associated sections in the main text.

1) **Circuit models and theoretical analysis**: is presented in Sec. 2 in the main text and the detailed introduction and rationale are presented in Appendix Sec. C
2) **1D NG Langevin sampling**: is presented in Sec. 4 in the main text and the detailed calculations are in Appendix Sec. C.
3) **Multivariate NG Langevin in coupled circuits**: is presented in Sec. 5 in the main text and the detailed calculations are in Appendix Sec. D.
4) **1D NG Hamiltonian sampling**: is presented in Sec. 6 in the main text and the detailed calculations are in Appendix Sec. E.
5) **Numerical simulation details**: is presented in Appendix Sec. F including the parameters for each figure. The complete code of simulation is provided in the supplementary files with detailed usage instructions.
6) **Generalization** is presented in Appendix Sec.G including the non-flat prior and Von Mises case.

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

APPENDIX

# A APPENDIX FIGURES

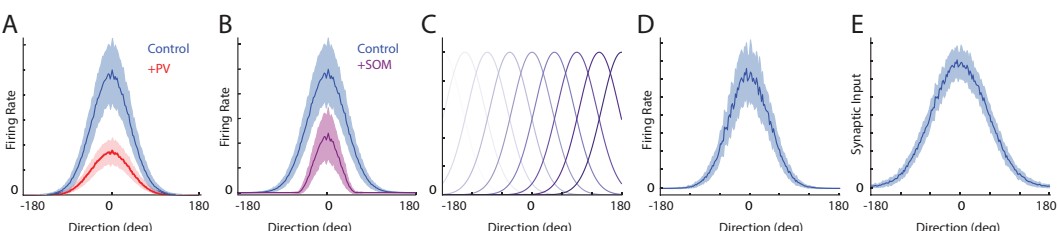

Figure A1: Supplementary figure of Fig. 1. (A-B)The tuning curve of an example E neuron in control state compared with enhancing PV neurons (A) and SOM neurons (B). It shows the PV neurons provide divisive inhibition to the E neuron, while the SOM provides subtractive inhibition to the E neuron. (C) The tuning curves of all E neurons in the circuit tile the whole stimulus feature space z. (D-E) The temporally averaged Gaussian profile of the firing rate $r_E(\theta)$ (D) and synaptic input $u_E(\theta)$ (E), supporting the Gaussian ansatz of the attractor states in Eqs. (2)

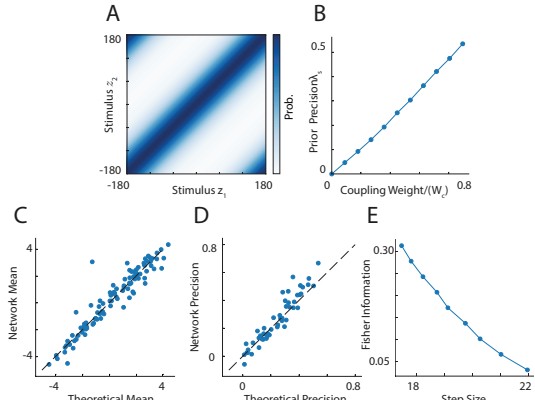

Figure A2: Supplementary figure of Fig. 3. (A) The joint correlational prior of the stimulus $z_1$ and $z_2$ stored in the coupled circuits presented in Fig. 3. The correlation between two stimuli is determined by width of the diagonal band . (B) The prior precision $\lambda_s$ increases with the coupling weight between two circuits. (C-D) The coupled circuits sample the posterior by using its internal subjective prior. Comparison of the sampling mean (C) and the prior precision (D) stored in the network with theoretical predictions. Each point represents results from a random combination of feedforward inputs, connection weights. (D) The picture shows the off-diagonal term of prior precision i.e. the joint part of posterior. (E) The sampling step size of coupled network changes with posterior FI

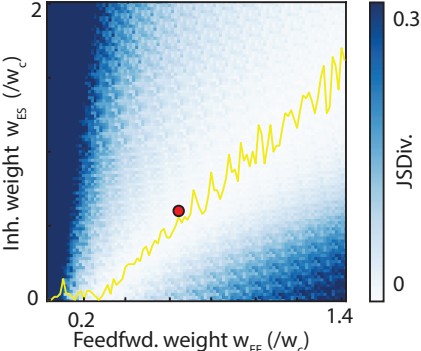

Figure A3: Supplementary figure of Fig. 4. The augmented circuit with E, PV and SOM neurons have a line manifold in the parameter space to sample posteriors correctly, suggesting no fine-tuning is needed. The parameter space is spanned by feedforward weight $w_{EF}$ and the inhibitory weight from SOM to E neurons $w_{ES}$. The red dot shows the network parameters we use for simulation

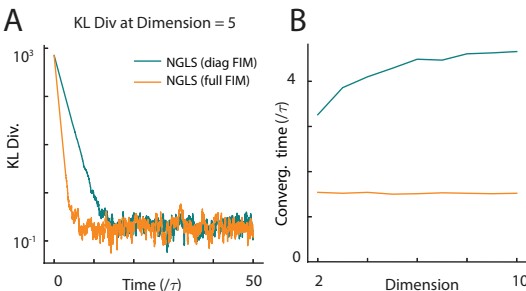

Figure A4: Supplementary figure of Fig. 3. Convergence time for multivariate posterior sampling coupled networks with the posterior dimensions. (A) At a dimension of 5, Natural gradient Langevin sampling with a diagonal Fisher information matrix converges more slowly than the full Fisher variant. (B) To determine the convergence time constant across dimensions, we fitted an exponential function to the KL divergence trajectories, excluding the noisy tails. These simulation results demonstrate that the full Fisher method maintains a consistent convergence time constant across dimensions, whereas the diagonal method is slower, whose convergence times increase with the posterior dimension.

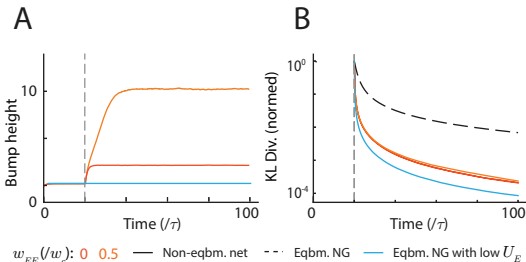

Figure A5: Supplementary figure of Fig. 2J-K. Non-equilibrium sampling in neural circuits (red and orange lines) whose sampling time constant $U_E$ gradually increases (panel A) compared with equilibrium sampling of simulating the bump position dynamics (Eq. 3a) but with deliberately clamped lower $U_E$ (blue line, panel A). Although the latter equilibrium bump position sampling (blue) is faster than the former non-equilibrium circuit sampling (red and orange), the latter is **infeasible** in neural circuit dynamics. This is because the circuit dynamics is automatic and has its intrinsic way to evolve the $U_E$ that lives in a subspace of the population activity, and it is infeasible to clamp $U_E$ in the circuit dynamics without affecting other subspaces.

## B  NATURAL GRADIENT LANGEVIN SAMPLING

### B.1  LANGEVIN DYNAMICS

The dynamics of Langevin sampling performs stochastic gradient ascent on the manifold of the log-posterior of stimulus features (Welling & Teh, 2011), which is written as,

$$\dot{\mathbf{z}}_t = \tau_L^{-1} \nabla \ln p(\mathbf{z}_t | \mathbf{r}_F) + (2\tau_L^{-1})^{1/2} \boldsymbol{\xi}_t, \quad (\nabla \equiv d/dz) \tag{B1}$$

where $\boldsymbol{\xi}_t$ is a multivariate independent Gaussian-white noise, satisfying $\langle \boldsymbol{\xi}_t \boldsymbol{\xi}_{t'}^\top \rangle = \mathbf{I}\delta(t - t')$, with $\mathbf{I}$ the identity matrix and $\delta(t - t')$ the Dirac delta function, and

$\tau_L$ is a positive-definite matrix (or a positive scalar in the 1D case) determining the sampling time constant, which is also called the pre-conditioning matrix. Importantly, $\tau_L$ is a free parameter of the sampling in that it doesn't change the equilibrium distribution of $\mathbf{z}_t$.

### B.2  NATURAL GRADIENT SAMPLING VIA FISHER INFORMATION (MATRIX)

Amari proposed the natural gradient method to utilize the geometry of the distribution to adaptively determine the sampling time constant $\tau_L$ that controls the sampling step size (Amari, 1998). Intuitively, for a widely spread distribution, we should choose a small time constant (large step size) that can speed up the convergence of the sampling. Vice versa, for a narrowly distributed latent variable, a

large time constant (small step size) is favoured to avoid instability of the sampling. Specifically, Amari's natural gradient method proposed the sampling time constant can be determined by using the Fisher information that is a measure of the local curvature of the distribution. In the framework of information geometry (Amari, 2016), the Fisher information matrix serves as a Riemannian metric on the statistical manifold of $\mathbf{z}$. Consider two neighboring (posterior) distributions $\pi(\mathbf{z})$ and $\pi(\mathbf{z} + \boldsymbol{d})$ with an infinitesimal displacement $\boldsymbol{d}$, a second-order Taylor series approximation reveals the Fisher information as the underlying distance metric.

$$D_{KL}\left[\pi(\mathbf{z}) \| \pi(\mathbf{z} + \boldsymbol{d})\right] \approx \frac{1}{2}\boldsymbol{d}^\top \mathbf{G}(\mathbf{z})\boldsymbol{d}$$

While the Fisher Information is often introduced in the context of the likelihood function in frequentist statistics, its definition can be generalized. For any probability distribution, its Fisher Information matrix measures the expected curvature of its logarithm. For the posterior $\pi(\mathbf{z}) \equiv p(\mathbf{z}|\mathbf{r}_F)$ to be sampled, we get the posterior information matrix (or Bayesian Fisher Information)(Amari, 2016),

$$G(z) = -\mathbb{E}_{p(\mathbf{r}_F|z)}\left[\nabla_\mathbf{z} \log \pi(\mathbf{z})\nabla_\mathbf{z} \log \pi(\mathbf{z})^\top\right]. \tag{B2}$$

It is symmetric and positive semi-definite. Then the posterior information matrix acts as a precondition to set up the time constant of the sampling(Girolami & Calderhead, 2011):

$$\dot{\mathbf{z}}_t = \tau_L^{-1}\nabla \ln \pi(\mathbf{z}) + (2\tau_L^{-1})^{1/2}\boldsymbol{\xi}_t, \quad \tau_L = \eta[\mathbf{G}(\mathbf{z}) + \alpha] \tag{B3}$$

Here, the time constant increases with $\mathbf{G}(\mathbf{z})$, which ensures a smaller step size (larger time constant) when the posterior is more curved (larger Fisher information). This adaptation improves sampling efficiency, as it accounts for anisotropies in the posterior, preventing slow mixing along directions of low curvature. The $\alpha$ is a regularization term that increases the numerical stability when inverting the time constant with a very small Fisher information $\mathbf{G}(\mathbf{z})$.

With more details, the Fisher information is the expected value of the negative Hessian matrix. It represents the curvature of the posterior on the statistical manifold where the latent variable $\mathbf{z}$ reside.

$$\mathbf{G}(\mathbf{z}) = -\mathbb{E}_{p(\mathbf{r}_F|z)}\left[\nabla_\mathbf{z}^2 \log \pi(\mathbf{z})\right] \tag{B4}$$

In many practical applications, a "flat" or "non-informative" prior is used for some or all parameters. The posterior information matrix simplifies to become identical to the likelihood's Fisher information matrix. If prior is flat, this metric tensor of posterior manifold becomes,

$$\mathbf{G}(\mathbf{z}) = -\mathbb{E}_{p(\mathbf{r}_F|z)}(p(\mathbf{r}_\mathbf{F}|z)) - \mathbb{E}_{p(\mathbf{r}_F|z)}(p(z)) = -\mathbb{E}_{p(\mathbf{r}_F|z)}(p(\mathbf{r}_\mathbf{F}|z))$$

### B.3 SAMPLING SPEED MEASURED BY THE DECAYING SPEED OF KL DIVERGENCE

It has been proved that the upper-bound of the KL-divergence between the distribution of sample $p_t(z) = T^{-1}\sum_t \delta(z - z_t)$ and the equilibrium distribution $p_\infty(z)$ decreases exponentially (Dong et al., 2022), i.e.,

$$D_{KL}\left[p_t(\mathbf{z}) \| p_\infty(\mathbf{z})\right] \leq D_{KL}\left[p_0(\mathbf{z}) \| p_\infty(\mathbf{z})\right]\exp(-ht)$$

where $p_0(\mathbf{z})$ denotes the initial distribution at $t = 0$, and $h$ denotes the smallest real-part of all eigenvalues of the drift matrix.

## C A SINGLE CANONICAL CIRCUIT AND 1D NATURAL GRADIENT SAMPLING: THEORY

We present the math of theoretical analyses of the reduced recurrent circuit model consisting of E and PV neurons based on continuous attractor network dynamics.

### C.1 CONTINUOUS ATTRACTOR NETWORK DYNAMICS

To simplify the reading, we copy the network dynamics of E neurons (Eq. 1a),

$$\tau \frac{\partial \mathbf{u}_E(\theta, t)}{\partial t} = -\mathbf{u}_E(\theta, t) + \rho \sum_{X=E,F}(\mathbf{W}_{EX} * \mathbf{r}_X)(\theta, t) + \sqrt{\tau\mathsf{F}[\mathbf{u}_E(\theta, t)]_+}\xi(\theta, t), \tag{C1}$$

and the divisive normalization provided by PV neurons (Eq. 1b),

$$\mathbf{r}_E(\theta) = \frac{[\mathbf{u}_E(\theta)]_+^2}{1 + \rho w_{EP} \int_{-\pi}^{\pi} [\mathbf{u}_E(\theta)]_+^2 d\theta}, \tag{C2}$$

and the recurrent connection kernel $\mathbf{W}_{EX}$ (Eq. 1d)

$$\mathbf{W}_{YX}(\theta) = w_{YX} \left(\sqrt{2\pi} a_{XY}\right)^{-1} \exp(-\theta^2/2a_{XY}^2). \tag{C3}$$

## C.2 Network's attractor states

We verify the proposed Gaussian ansatz of the attractor states of E neurons (Eq. 2),

$$\bar{\mathbf{u}}_E(\theta) = \bar{U}_E \exp\left[-\frac{(\theta - z_E)^2}{4a_E^2}\right]. \tag{C4}$$

First, we substitute it into the divisive normalization (Eq. C2), yielding the following expression for the firing rate of E neurons,

$$\bar{\mathbf{r}}_E(\theta) = \frac{[\bar{\mathbf{u}}_E^2(\theta)]_+^2}{1 + \rho w_{EP} \int [\bar{\mathbf{u}}_E(\theta)]_+^2 d\theta} = \underbrace{\frac{\bar{U}_E^2}{1 + \rho w_{EP} \bar{U}_E^2 \sqrt{2\pi} a_E}}_{\bar{R}_E} \exp\left[-\frac{(\theta - z_E)^2}{2a_E^2}\right]. \tag{C5}$$

Then we use the above E firing rate (Eq. C5) to calculate the recurrent input from the neuronal population of type $Y$ to the one with type $X$ in the circuit model,

$$\begin{aligned}
\mathbf{u}_{XY}(\theta) &= \rho \mathbf{W}_{XY} * \mathbf{r}_Y(\theta) \\
&= \frac{\rho w_{XY} R_Y}{\sqrt{2\pi} a_{XY}} \int \exp\left[-\frac{(\theta' - \theta)^2}{2a_{XY}^2} - \frac{(\theta' - z_Y)^2}{2a_Y^2}\right] d\theta' \\
&= \rho w_{XY} R_Y \frac{a_Y}{\sqrt{a_{XY}^2 + a_Y^2}} \exp\left[-\frac{(\theta - z_Y)^2}{2(a_{XY}^2 + a_Y^2)}\right].
\end{aligned} \tag{C6}$$

Specifically, based on Eq. (C6), the recurrent E population input is

$$\mathbf{u}_{EE}(\theta) = \rho \mathbf{W}_{EE} * \mathbf{r}_E(\theta) = \underbrace{\frac{\rho}{\sqrt{2}} w_{EE} R_E}_{U_{EE}} \exp\left[-\frac{(\theta - z_E)^2}{4a_E^2}\right], \tag{C7}$$

and the feedforward population input is,

$$\mathbf{u}_{EF}(\theta) = \rho \mathbf{W}_{EF} * r_F(\theta) = \underbrace{\frac{\rho}{\sqrt{2}} w_{EF} R_F}_{U_{EF}} \exp\left[-\frac{(\theta - \mu_z)^2}{4a_E^2}\right]. \tag{C8}$$

It can be checked the proposed Gausian ansatz (Eq. C4) is indeed the sum of the recurrent input (Eq. C7) and the feedforward input (Eq. C8), i.e.,

$$\bar{\mathbf{u}}_E(\theta) = \mathbf{u}_{EE}(\theta) + \mathbf{u}_{EF}(\theta)$$

$$\bar{U}_E \exp\left[-\frac{(\theta - z_E)^2}{4a_E^2}\right] = U_{EE} \exp\left[-\frac{(\theta - z_E)^2}{4a_E^2}\right] + U_{EF} \exp\left[-\frac{(\theta - \mu_z)^2}{4a_E^2}\right]$$

and implies

$$\mathbf{r}_E(\theta, t) = U_{EE} + U_{EF}, \quad z_E = \mu_z.$$

This completes the recurrent loop of the dynamics, and verify the validity of the Gaussian ansatz (Eq. 2).

### C.3 DIMENSIONALITY REDUCTION BY PROJECTING ON DOMINANT MODES

We substitute Eqs. (C4-C8) into the Eq. (C1),

$$
\tau \dot{U}_E \exp\left[-\frac{(\theta - z_E)^2}{4a_E^2}\right] + \frac{\tau U_E}{2a_E} \dot{z}_E \left(\frac{\theta - z_E}{a_E}\right) \exp\left[-\frac{(\theta - z_E)^2}{4a_E^2}\right]
$$

$$
= -U_E \exp\left[-\frac{(\theta - z_E)^2}{4a_E^2}\right] + \frac{\rho}{\sqrt{2}} w_{EE} R_E \exp\left[-\frac{(\theta - z_E)^2}{4a_E^2}\right] \tag{C9}
$$

$$
+ \frac{\rho}{\sqrt{2}} w_{EF} R_F \exp\left[-\frac{(\theta - \mu)^2}{4a_E^2}\right] + \sqrt{\tau \mathsf{F} U_E} \exp\left[-\frac{(\theta - z_E)^2}{8a_E^2}\right] \xi(\theta, t)
$$

Previous studies analytically calculated the first two dominant eigenvectors (modes) (Wu et al., 2016; Fung et al., 2010), corresponding to the change of the position and height of the Gaussian ansatz respectively,

$$
\text{Position}: \quad \phi_1(\theta|z_E) \propto \nabla_z \bar{\mathbf{u}}_E(\theta) \propto (\theta - z_E) \exp[-(\theta - z_E)^2/4a^2], \tag{C10a}
$$

$$
\text{Height}: \quad \phi_2(\theta|z_E) \propto \quad \bar{\mathbf{u}}_E(\theta) \propto \exp[-(\theta - z_E)^2/4a^2]. \tag{C10b}
$$

Projecting the dynamics Eq. (C9) into these 2 motion modes (Eq. C10), which means calculate the inner product $\int f(\theta)\phi(\theta|z_E)d\theta$ with $f(\theta)$ a term in Eq. (C9),

$$
\tau U_E \dot{z}_E = \frac{\rho}{\sqrt{2}} w_{EF} R_F (\mu - z_E) \exp\left[-\frac{(\mu - z_E)^2}{8a_E^2}\right] + \sigma_z \sqrt{\tau U_E} \xi
$$

$$
\tau \dot{U}_E = -U_E + \frac{\rho}{\sqrt{2}} w_{EE} R_E + \frac{\rho}{\sqrt{2}} w_{EF} R_F \exp\left[-\frac{(\mu - z_E)^2}{8a_E^2}\right] + \sigma_U \sqrt{\tau U_E} \xi
$$

where

$$
\sigma_z^2 = \frac{8a\mathsf{F}}{3\sqrt{3\pi}}, \quad \sigma_U^2 = \frac{\mathsf{F}}{\sqrt{3\pi}a}. \tag{C11}
$$

When the bump position $z_E$ is near the input position, i.e., $\mu - z_E \ll a_E$, which is usually the case in the circuit model, the exponential term $\exp[-(\mu - z_E)^2/8a_E^2]$ is close to one and can be safely ignored,

$$
\dot{z}_E = (\tau U_E)^{-1} \frac{\rho}{\sqrt{2}} w_{EF} R_F (\mu_z - z_E) + \sigma_z (\tau U_E)^{-1/2} \xi_t \tag{C12}
$$

$$
\dot{U}_E = \tau^{-1}[-U_E + \frac{\rho}{\sqrt{2}} (w_{EE} R_E + w_{EF} R_F)] + \sigma_U (\tau^{-1} U_E)^{1/2} \xi_t. \tag{C13}
$$

Furthermore, by using the notation

$$
U_{EF} = \rho w_{EF} R_F / \sqrt{2}, \quad U_{EE} = \rho w_{EE} R_E / \sqrt{2},
$$

we arrive at Eqs. (3a and 3b) in the main text.

### C.4 THE PROBABILISTIC GENERATIVE MODEL EMBEDDED IN THE CIRCUIT MODEL

#### C.4.1 THE STIMULUS LIKELIHOOD

We study how feedforward input defines the latent stimulus likelihood, i.e., $\mathcal{L}(z) \propto p(\mathbf{r}_F|z)$. From the Eq. (1e), the feedforward input $\mathbf{r}_F$ is modeled as a set of independent Poisson spike trains, where each neuron's firing rate is Gaussian-tuned to the stimulus (Ma et al., 2006):

$$
\mathbf{r}_F(\theta|z) \sim \text{Poisson}[\lambda_F(\theta|z)], \quad \lambda_F(\theta|z) = R_F \exp[-(\theta - z)^2/2a^2], \tag{C14}
$$

where $\lambda_F(\theta|z)$ is the mean firing rate of the neuron with stimulus preference $\theta$. $\mathbf{r}_F$ denotes the peak input rate, and $a$ specifies the tuning width. Explicitly writing the Poisson distribution of feedforward input spikes (we discretize the continuous $\theta$ into equally spaced $\theta_j$),

$$
p(\mathbf{r}|z) = \prod_{j=1}^{N_E} \text{Poisson}\left(\mathbf{r}_j|\lambda_j \Delta t\right) = \prod_{j=1}^{N_E} \frac{(\lambda_j \Delta t)^{\mathbf{r}_j}}{\mathbf{r}_j!} \exp(-\lambda_j \Delta t). \tag{C15}
$$

Taking the logarithm,

$$\ln p(\mathbf{r}|z) = \sum_j \left[ \mathbf{r}_j \ln(\lambda_j \Delta t) - \ln(\mathbf{r}_j!) - \lambda_j \right],$$
$$= \sum_j \mathbf{r}_j \ln(\lambda_j \Delta t) + \text{const.} \tag{C16}$$

The const. in the above equation is under the assumption that the sum of population firing rate $\sum_j \lambda_j$ is a constant irrelevant to latent stimulus $z$, which is true in a homogeneous population with a large number of neurons. Substituting the expression of the Gaussian tuning,

$$\ln p(\mathbf{r}|z) = -\sum_j \mathbf{r}_j \frac{(\theta - z)^2}{2a^2} + \text{const} = -\frac{1}{2}\Lambda(z - \mu_z)^2 + \text{const}, \tag{C17}$$

where

$$\mu_z = \frac{\sum_j \mathbf{r}(\theta_j)\theta_j}{\sum_j \mathbf{r}(\theta_j)}, \quad \Lambda = a^{-2} \sum_j \mathbf{r}(\theta_j) \approx \sqrt{2\pi}\rho a^{-1} R_F. \tag{C18}$$

This implies the latent stimulus likelihood for the latent stimulus feature $z$ given an observed feedforward input $\mathbf{r}_F$ is derived as a Gaussian distribution,

$$\mathcal{L}(z) = \mathcal{N}(z|\mu_z, \Lambda^{-1}),$$

which is the Eq. (8) in the main text. Notably, the Gaussian distribution comes from the profile of the Gaussian tuning (Eq. C14) (Ma et al., 2006).

### C.4.2 UNIFORM STIMULUS PRIOR IN THE CIRCUIT

Comparing the E bump position dynamics (Eq. C12) with the Langevin sampling dynamics (Eq. B3), it immediately suggests that the circuit stores a uniform (uninformative) stimulus prior, i.e., $p(z)$ is uniform. This is because the gradient of the log-likelihood ($\nabla \mathcal{L}(z)$, Eq. 8) has the same form with the drift term in the E position dynamics

$$\text{Likelihood gradient:} \qquad \nabla \ln \mathcal{L}(z) = \Lambda(\mu_z - z)$$
$$\text{E bump position drift term:} \quad U_{EF}(\mu_z - z_E)$$

suggesting the gradient of the prior is zero, i.e., $\nabla \ln p(z) = 0$. This uniform prior arises from the circuit's homogeneous neurons (uniformly distributed in feature space) and its translation-invariant connection profile. Consequently, for the circuit to store a non-uniform prior, it must break this inherent symmetry in its neural organization and connectivity.

### C.5 CONDITIONS FOR REALIZING LANGEVIN SAMPLING IN THE CIRCUIT

The circuit sampling of the likelihood means the equilibrium distribution of the bump position (Eq. C12) should match with the likelihood (Eq. 8). We copy the circuit bump position dynamics and the likelihood Langevin sampling dynamics in below for comparison,

$$\text{Circuit:} \qquad \dot{z}_E = (\tau U_E)^{-1} \underbrace{\frac{\rho w_{EF} R_F}{\sqrt{2\Lambda}}}_{\lambda_z} \Lambda(\mu_z - z_E) + \sigma_z(\tau U_E)^{-1/2}\xi_t,$$

$$\text{Langevin:} \quad \dot{z}_t = \tau_L^{-1}\Lambda(\mu_z - z) + (2\tau_L^{-1})^{1/2}\xi_t.$$

The $\sigma_z$ is a constant that doesn't change with neuronal activities. Therefore, the likelihood Langevin sampling in the circuit can be realized by setting the feedforward weight $w_{EF}$ appropriately to make the ratio of the drift and diffusion coefficients the same as the Langevin sampling dynamics. The optimal feedforward weight can be found as (by using Eq. C18)

$$\frac{\sigma_z^2}{\lambda_z} = 2 \quad \Leftrightarrow \quad w_{EF} = \frac{\sigma_z^2}{\sqrt{2}\rho} \frac{\Lambda}{R_F} = \left(\frac{2}{\sqrt{3}}\right)^3 \mathsf{F} \tag{C19}$$

Furthermore, the time constant of the $z_E$ dynamics is

$$\tau_z = \lambda_z^{-1}\tau U_E = \frac{2\sqrt{\pi}}{a w_{EF}}\tau U_E, \tag{C20}$$

which is proportional to the E bump height $U_E$. Finally, the equation of bump position (Eq. C12) can be converted into the same form with a standard Langevin sampling,

$$\dot{z}_E = \tau_z^{-1}\Lambda(\mu - z_E) + (2\tau_z^{-1})^{1/2}\xi_t$$

### C.6 NATURAL GRADIENT SAMPLING IN THE CIRCUIT

The natural gradient Langevin sampling utilizes the Fisher information to determine the sampling time constant (Eq. B3). We verify whether this can be realized in the circuit dynamics. Firstly, the Fisher information of the likelihood is (Eqs. B4 and C17),

$$
\begin{aligned}
G(z) &= -\mathbb{E}\left[\nabla^2 \log \mathcal{L}(z)\right], \quad \text{where} \quad \mathcal{L}(z) = \mathcal{N}(z|\mu_z, \Lambda^{-1}), \\
&= \Lambda \\
&= \sqrt{2\pi}\rho a^{-1} R_F
\end{aligned}
\tag{C21}
$$

Meanwhile, the time constant of the circuit sampling dynamics $\tau_z$ is proportional to the bump height $U_E$ (Eq. C20). From the Eq. (C13), the equilibrium mean of the bump height can be calculated as

$$
\bar{U}_E = \underbrace{\frac{\rho}{\sqrt{2}}w_{EE}\bar{R}_E}_{U_{EE}} + \underbrace{\frac{\rho}{\sqrt{2}}w_{EF}R_F}_{U_{EF}} = U_{EE} + \underbrace{\frac{aw_{EF}}{2\sqrt{\pi}}}_{\lambda_z} G(z).
\tag{C22}
$$

And therefore the circuit's sampling time constant is

$$
\text{Circuit:} \qquad \tau_z = \lambda_z^{-1}\tau U_E = \tau\left[G(z) + \lambda_z^{-1}U_{EE}\right],
\tag{C23}
$$

$$
\text{Natural gradient:} \quad \tau_L = \eta[G(z) + \alpha]
\tag{C24}
$$

It clearly shows the bump height $\bar{U}_E$ increases with the Fisher information $G(z)$. Moreover, the recurrent E input $U_{EE}$ acts as the regularization term to increase the numerical stability of inverting the Fisher information (similar to the role of $\alpha$ in Eq. B3). This proves the reduced circuit with E and PV neurons indeed implements natural gradient Langevin sampling from the likelihood.

## D COUPLED NEURAL CIRCUITS AND MULTIVARIATE POSTERIOR SAMPLING: THEORY

### D.1 THEORETICAL ANALYSIS OF THE COUPLED CIRCUIT DYNAMICS

We present the math about coupled canonical neural circuits implementing multivariate stimulus posterior inference via natural gradient Langevin sampling (Zhang et al., 2016; 2023; Raju & Pitkow, 2016). The model we consider is composed of $M$ reciprocally connected coupled circuit, with each the same as a single canonical circuit in Sec. C. Each circuit $m$ receives a feedforward input independently generated from the corresponding latent stimulus $s_m$ (Fig. 3), and eventually draw the stimulus $z_m$ from the multivariate posterior. Therefore, the number of coupled circuits in the model is determined by the dimension of the multivariate posteriors.

The dynamics of the coupled circuits is written as (we raise the subscript of capital latter denoting neuron and input types to the superscript, and the new subscripts of lowercase letters denote the E population indices),

$$
\tau\frac{\partial \mathbf{u}_m^E(\theta, t)}{\partial t} = -\mathbf{u}_m^E(\theta, t) + \rho \sum_{X=E,F} \sum_{n=1}^{M}(\mathbf{W}_{mn}^{EX} * \mathbf{r}_n^X)(\theta, t) + \sqrt{\tau\mathsf{F}[\mathbf{u}_m^E(\theta, t)]_+}\xi_m(\theta, t)
\tag{D1}
$$

Each circuit $\mathbf{u}_m^E(\theta)$ receives a feedforward input $\mathbf{r}_m^F(\theta)$ that is independently generated from a latent stimulus $s_m$ via the same way in the single circuit (Fig. 3, Eq. C14),

$$
\mathbf{r}_m^F(\theta|z) \sim \text{Poisson}[\lambda_m^F(\theta|z_m)], \quad \lambda_m^F(\theta|z_m) = R_m^F \exp[-(\theta - z_m)^2/2a^2],
$$

For simplicity, we consider the feedforward connection weight $w_{mm}^{EF}$ of each circuit is the same.

Similar to the one-dimensional case (Eq. C4), we consider the Gaussian ansatz for the population synaptic input at each circuit $m$,

$$
\mathbf{u}_m^E(\theta, t) = \bar{U}_m^E(t)\exp\left[-\frac{(\theta - z_m^E)^2}{4a^2}\right].
$$

Performing similar calculations by substituting the Gaussian ansatz of each circuit into the dynamics of the coupled circuits (Eq. D1),

$$\tau \frac{U_{E,m}}{2a} \frac{dz_{mt}}{dt} \frac{\theta - z_{mt}}{a} e^{-(\theta - z_{mt})^2/4a^2} + \frac{\tau}{2a} \frac{dU_{E,m}}{dt} e^{-(\theta - z_{mt})^2/4a^2},$$

$$= -U_{E,m} e^{-(\theta - z_{mt})^2/4a^2} + \frac{\rho}{\sqrt{2}} \sum_n w_{mn}^{EE} R_n^E e^{-(\theta - z_{nt})^2/4a^2}, \tag{D2}$$

$$+ \frac{\rho}{\sqrt{2}} w_{mm}^{EF} R_m^F e^{-(\theta - \mu_m)^2/4a^2} + \sqrt{\tau F U_m^E} e^{-(\theta - z_{mt})^2/8a^2} \xi_{mt}.$$

Projecting the above dynamics onto the two eigenfunctions (C10), and assume the differences between the bump positions of different circuits are small enough compared with the tuning width $a$, i.e., $|z_n - z_m| \ll a$,

Position: $\quad \dfrac{dz_{mt}}{dt} = \dfrac{\rho}{\sqrt{2}} \left(\tau U_m^E\right)^{-1} \left[ \sum_n w_{mn}^{EE} R_n^E (z_{nt} - z_{mt}) + w_{mm}^{EF} R_m^F (\mu_m - z_{mt}) \right]$

$$+ \sigma_z \left(\tau U_m^E\right)^{-1/2} \xi_{mt}$$

Height: $\quad \tau \dfrac{dU_m^E}{dt} = -U_m^E + \dfrac{\rho}{\sqrt{2}} \sum_n w_{mn}^{EE} R_n^E + \dfrac{\rho}{\sqrt{2}} w_{mm}^{EF} R_m^F + \sigma_U \left(\tau U_m^E\right)^{1/2} \xi_{mt}.$

where $\sigma_z$ and $\sigma_U$ are the same as Eq. (C11). Reorganizing the above equation into the matrix form,

Position: $\quad \dot{\mathbf{z}}_E = (\tau \mathbf{D_U})^{-1} \left[ -\mathbf{L}\mathbf{z}_E + \mathbf{U}_{EF} \circ (\boldsymbol{\mu} - \mathbf{z}_E) \right] + \sigma_z (\tau \mathbf{D_U})^{-1/2} \xi_t, \tag{D3a}$

Height: $\quad \dot{\mathbf{U}}_E = \tau^{-1} \left( -\mathbf{U}_E + \mathbf{U}_{EE} + \mathbf{U}_{EF} \right) + \sigma_U (\tau^{-1} \mathbf{D_U})^{1/2} \xi_t. \tag{D3b}$

where $\circ$ denotes the element-wise multiplication, and

$$\mathbf{U}_E = \{U_m^E\}_{m=1}^M, \quad \mathbf{z}_E = \{z_m\}_{m=1}^M,$$

$$\mathbf{U}_{EE} = \{U_m^{EE}\}_{m=1}^M, \quad \text{with} \quad U_m^{EE} = \sum_n U_{mn}^{EE} = \sum_n \frac{\rho}{\sqrt{2}} w_{mn}^{EE} R_n^E,$$

$$\mathbf{U}_{EF} = \{U_m^{EF}\}_{m=1}^M, \quad \text{with} \quad U_m^{EF} = \frac{\rho}{\sqrt{2}} w_{mm}^{EF} R_m^F, \tag{D4}$$

$$\text{Matrix } \mathbf{L}: \quad [\mathbf{L}]_{mn} = -U_{mn}^{EE} \ (m \neq n), \quad \text{and} \quad [\mathbf{L}]_{mm} = -\sum_{n \neq m} [\mathbf{L}]_{mn},$$

$$\text{Matrix } \mathbf{D_U} = \text{diag}(\mathbf{U}_E)$$

We obtain the bump position and height dynamics embedded in neural dynamics as presented in Eqs. (12a-12b) in the main text.

## D.2 THE GENERATIVE MODEL OF MULTIVARIATE STIMULUS STORED IN THE CIRCUIT

We present the math analysis in identifying the generative model especially the subjective stimulus prior stored in the circuit. Generally, the multivariate stimulus posteriors given received feedforward inputs are,

$$\pi(\mathbf{z}) \equiv p\big(\mathbf{z} | \{\mathbf{r}_m^F\}_{m=1}^M\big)$$

$$\propto p\big(\{\mathbf{r}_m^F\}_{m=1}^M | \mathbf{z}\big) p(\mathbf{z})$$

$$= \big[ \textstyle\prod_{m=1}^M p\big(\mathbf{r}_m^F | z_m\big) \big] p(\mathbf{z})$$

$$= \big[ \textstyle\prod_{m=1}^M \mathcal{N}(z_m | \mu_m, \Lambda_m^{-1}) \big] p(\mathbf{z}),$$

$$= \mathcal{N}(\mathbf{z} | \boldsymbol{\mu}, \boldsymbol{\Lambda}) p(\mathbf{z})$$

where the second last equality comes from by using the same derivations as the Sec. C.4.1 on each feedforward input $\mathbf{r}_m^F$. And

$$\boldsymbol{\Lambda} = \text{diag}(\Lambda_1, \Lambda_2, \cdots, \Lambda_M), \quad \text{where} \quad \Lambda_m = \sqrt{2\pi} \rho a^{-1} R_m^F$$

is the likelihood precision matrix. Note that the stimulus prior $p(\mathbf{z})$ is still unspecified at this moment. We will determine it in the following.

**Subjective prior stored in the coupled circuits**

Utilizing the Langevin sampling dynamics to sample the posterior

$$\dot{\mathbf{z}}_t = \tau_L^{-1} \nabla \ln \pi(\mathbf{z}) + (2\tau_L^{-1})^{1/2} \boldsymbol{\xi}_t,$$
$$= \tau_L^{-1} [\nabla \ln p(\mathbf{z}) + \boldsymbol{\Lambda} \circ (\boldsymbol{\mu} - \mathbf{z})] + (2\tau_L^{-1})^{1/2} \boldsymbol{\xi}_t,$$

Meanwhile, the coupled circuits' bump position dynamics is

$$\dot{\mathbf{z}}_E = (\tau \mathbf{D_U})^{-1} \big[ -\mathbf{L}\mathbf{z}_E + \mathbf{U}_{EF} \circ (\boldsymbol{\mu} - \mathbf{z}_E) \big] + \sigma_z (\tau \mathbf{D_U})^{-1/2} \xi_t,$$

Using the definition of $\mathbf{U}_{EF}$ (Eq. D4) and the feedforward input intensity with the likelihood precision (Eq. C18),

$$\mathbf{U}_{EF} = \underbrace{\frac{w_{mm}^{EF} a}{2\sqrt{\pi}}}_{\lambda_z} \boldsymbol{\Lambda}$$

It is straightforward to regard the $\mathbf{L}\mathbf{z}$ term as the gradient from the stimulus prior,

$$\nabla \ln p(\mathbf{z}) = -\lambda_z^{-1} \mathbf{L}\mathbf{z} \quad \Leftrightarrow \quad p(\mathbf{z}) \propto \exp(-\mathbf{z}^\top \mathbf{L}\mathbf{z}/2\lambda_z) \tag{D5}$$

Specifically, the prior precision matrix $\lambda_z^{-1}\mathbf{L}$ is a generalized Laplacian matrix (Eq. D4, whose determinant is zero, i.e., $|\mathbf{L}| = 0$, suggesting the marginal prior of each stimulus is uniform, i.e., $p(z_m)$ is uniform. As an example, for $M = 2$, the prior $p(\mathbf{z} = (z_1, z_2)^\top)$ is written as,

$$p(\mathbf{z}) = \exp\left[ -\frac{\mathbf{L}_{12}}{2} \mathbf{z}^\top \begin{pmatrix} 1 & -1 \\ -1 & 1 \end{pmatrix} \mathbf{z} \right] = \exp\left[ -\frac{\mathbf{L}_{12}}{2} (z_1 - z_2)^2 \right], \tag{D6}$$

where $\mathbf{L}_{12}$ characterizes the correlation between $z_1$ and $z_2$. It can be checked each marginal stimulus prior is uniform.

**Subjective multivariate stimulus posterior in the circuit**

Based on the identified stimulus prior stored in the circuit (Eq. D5), the (subjective) stimulus posterior is calculated as

$$\pi(\mathbf{z}) \equiv p\big(\mathbf{z} | \{\mathbf{r}_m^F\}_{m=1}^M\big)$$
$$= \mathcal{N}(\mathbf{z}|\boldsymbol{\mu}, \boldsymbol{\Lambda}) \mathcal{N}(\mathbf{z}|\mathbf{0}, \lambda_z \mathbf{L}^{-1})$$
$$\equiv \mathcal{N}(\mathbf{z}|\boldsymbol{\mu_z}, \boldsymbol{\Omega}^{-1})$$

where

$$\boldsymbol{\Omega} = \boldsymbol{\Lambda} + \lambda_z^{-1}\mathbf{L}, \quad \boldsymbol{\mu_z} = \boldsymbol{\Omega}^{-1}\boldsymbol{\Lambda}\boldsymbol{\mu}. \tag{D7}$$

### D.3 Natural gradient sampling via diagonal approximation of Fisher information matrix

Eq. (D3a) suggests the time constant of the circuit's sampling dynamics (bump position) is determined by the matrix $\mathbf{D_U}$.

$$\mathbf{D_U} = \mathrm{diag}(\bar{\mathbf{U}}_E) \quad \text{where} \quad \bar{\mathbf{U}}_E = \mathbf{U}_{EE} + \mathbf{U}_{EF}$$

We next analyze its relation with the Fisher information to verify whether the circuit implement natural gradient sampling for multivariate posteriors.

Fisher information of the multivariate stimulus

Based on the (subjective) multivariate posterior calculated by the circuits (Eq. D7), the Fisher information matrix of the multivariate stimulus is,

$$\mathbf{G}(\mathbf{z}) = \boldsymbol{\Omega} = \lambda_z^{-1}\mathbf{L} + \boldsymbol{\Lambda} = \lambda_z^{-1}[\mathbf{L} + \mathrm{diag}(\mathbf{U}_{EF})] \tag{D8}$$

In particular, by using the definition of the prior precision matrix (Eq. D5) and the posterior precision (Eq. D7),

$$\text{diag}(\mathbf{G}(\mathbf{z})) = \lambda_z^{-1}(\text{diag}(\mathbf{L}) + \mathbf{U}_{EF}),$$
$$= \lambda_z^{-1}[\text{diag}(\mathbf{U}_{EE}) + \text{diag}(\mathbf{U}_{EF})],$$
$$= \lambda_z^{-1}\text{diag}(\bar{\mathbf{U}}_E),$$

which clearly shows the circuit's sampling time constant $\mathbf{D_U}$ is the diagonal matrix of the full Fisher information matrix, giving rise to the Eq. (14) in the main text.

## E   NATURAL GRADIENT HAMILTONIAN SAMPLING IN THE CIRCUIT WITH SOM NEURONS

### E.1   CIRCUIT DYNAMICS

We also copy the dynamics of a single augmented circuit with SOM neurons (Eq. 1a and Eq. 1c) below.

$$\tau \dot{\mathbf{u}}_E(\theta, t) = -\mathbf{u}_E(\theta, t) + \rho \sum_{X=E,F,S} (\mathbf{W}_{EX} * \mathbf{r}_X)(\theta, t) + \sqrt{\tau \mathsf{F}[\mathbf{u}_E(\theta, t)]_+} \xi(\theta, t)$$
$$\tau \dot{\mathbf{u}}_S(\theta, t) = -\mathbf{u}_S(\theta, t) + \rho (\mathbf{W}_{SE} * \mathbf{r}_E)(\theta, t); \quad \mathbf{r}_S(\theta, t) = g_S \cdot [\mathbf{u}_S(\theta, t)]_+,$$

(E1)

Similar to the Gaussian ansatz presented in Eqs. (C4-C8), we also propose the same Gaussian ansatz for the synaptic inputs of E and SOM neurons respectively. Specifically, since SOM neurons have different activation function with the E neurons, the population firing rate of SOM neurons is,

$$\bar{\mathbf{r}}_S(\theta) = g_S \cdot \mathbf{u}_S(\theta, t) = \underbrace{g_S U_S}_{R_S} \exp\left[-\frac{(\theta - z_S)^2}{4a_S^2}\right].$$

(E2)

Substituting the Gaussian ansatz of E and SOM neurons into the circuit dynamics (Eqs. E1),

$$U_E \exp\left[-\frac{(\theta - z_E)^2}{4a_E^2}\right] = \frac{\rho}{\sqrt{2}}\left(w_{EE}R_E \exp\left[-\frac{(\theta - z_E)^2}{4a_E^2}\right] + w_{EF}R_F \exp\left[\frac{(\theta - \mu_z)^2}{4a_E^2}\right]\right.$$
$$\left. + \frac{\rho}{\sqrt{2}}w_{ES}R_S\frac{a_S}{a_E} \exp\left[-\frac{(\theta - z_S)^2}{4a_E^2}\right]\right),$$

(E3)

$$U_S \exp\left[-\frac{(\theta - z_S)^2}{4a_S^2}\right] = \rho w_{SE}R_E \frac{a_E}{\sqrt{a_{SE}^2 + a_E^2}} \exp\left[-\frac{(\theta - z_E)^2}{2(a_{SE}^2 + a_E^2)}\right],$$

Since the above equations are summations of Gaussian functions, it can be checked that when the positions of Gaussian functions are the same, i.e., $z_E = z_S = \mu_z$, the sum of two Gaussian functions will also be a Gaussian function. In addition, to validate the Gaussian ansatz, we need the width fulfilling the following constrain of the connection width,

$$2a_S^2 = a_{SE}^2 + a_E^2$$
$$a_E^2 = a_{ES}^2 + a_{SE}^2.$$

Similar to the two motion modes for E neuron, the SOM also have two motion nodes (Sale & Zhang, 2024),

$$\text{Position}: \quad \phi_1(\theta|z_S) \propto \nabla_z \bar{\mathbf{u}}_S(\theta) \propto (\theta - z_S)\exp[-(\theta - z_S)^2/4a_S^2], \quad \text{(E4a)}$$
$$\text{Height}: \quad \phi_2(\theta|z_S) \propto \quad \bar{\mathbf{u}}_S(\theta) \propto \exp[-(\theta - z_S)^2/4a_S^2]. \quad \text{(E4b)}$$

We project the dynamics of $\mathbf{u}_E$ and $\mathbf{u}_S$ onto their respective position modes (Eq. C10 and Eq. E4 respectively). From here, we assume the difference between neuronal populations' positions is small enough compared to the connection width $a$, i.e., $|z_E - z_S|$ and $|\mu_z - z_E| \ll 4a_X$. In this case, the projected circuit dynamics can be simplified by ignoring exponential terms in Eq. (E3),

$$\tau U_E \dot{z}_E = \frac{\rho}{\sqrt{2}}\left[w_{ES}R_S\frac{a_S}{a_E}(z_S - z_E) + w_{EF}R_F(\mu_z - z_E)\right] + \sigma_z\sqrt{\tau U_E}\eta_t$$
$$\tau U_S \dot{z}_S = \frac{\rho}{\sqrt{2}}\frac{a_E}{a_S}w_{SE}R_E(z_E - z_S)$$

(E5)

Similarly, we project the E and SOM's dynamics on their respective height modes,

$$\tau\dot{U}_E = -U_E + \frac{\rho}{\sqrt{2}}w_{EE}R_E + \frac{\rho}{\sqrt{2}}\frac{a_S}{a_E}w_{ES}R_S + \frac{\rho}{\sqrt{2}}w_{EF}R_F + \sigma_U\sqrt{\tau U_E}\xi_t \quad \text{(E6)}$$

$$\tau\dot{U}_S = -U_S + \frac{\rho}{\sqrt{2}}\frac{a_E}{a_S}w_{SE}R_E. \quad \text{(E7)}$$

Similarly, to simplify notations, we define

$$U_{XY} = \frac{\rho a_Y}{\sqrt{2}a_X}w_{XY}R_Y, \quad \text{(E8)}$$

and $\sigma_z$ and $\sigma_U$ are the same as Eq. (C11). The Eq. (E5) is simplified into,

$$\tau U_E \dot{z}_E = U_{ES}(z_S - z_E) + U_{EF}(\mu_z - z_E) + \sigma_z\sqrt{\tau U_E}\eta_t, \\ \tau U_S \dot{z}_S = U_{SE}(z_E - z_S), \quad \text{(E9)}$$

Reorganizing the bump position dynamcis into the matrix form,

$$\dot{\mathbf{z}} = (\tau\mathbf{D_U})^{-1}(\mathbf{F}_1\mathbf{z} + \mathbf{M}_1) + (\tau\mathbf{D_U})^{-1/2}\mathbf{\Sigma}_1\boldsymbol{\xi}_t \quad \text{(E10)}$$

where

$$\mathbf{z} = (z_E, z_S)^\top, \quad \mathbf{D_U} = \text{diag}(U_E, U_S),$$

$$\mathbf{F}_1 = \begin{pmatrix} -U_{EF} - U_{ES} & U_{ES} \\ U_{SE} & -U_{SE} \end{pmatrix}, \quad \mathbf{M}_1 = \begin{pmatrix} U_{EF}\mu_z \\ 0 \end{pmatrix}, \quad \mathbf{\Sigma}_1 = \begin{pmatrix} \sigma_z & 0 \\ 0 & 0 \end{pmatrix} \quad \text{(E11)}$$

### E.2 HAMILTONIAN SAMPLING IN THE CIRCUIT

In the present study, we consider a Hamiltonian sampling with friction, because it can be mapped to the proposed circuit with a diversity of interneurons. Hamiltonian sampling can sample the desired distribution $\pi(z)$ (with $\pi(z)$ as the equilibrium distribution), which is defined as,

$$\pi(z, p) = \exp[-H(z, p)] = \exp[-\ln\pi(z) - \ln\pi(p|z)] \quad \text{(E12)}$$

The previous study suggested the $z_E$ dynamics is a mixture of the Langevin sampling and the Hamiltonian sampling (Sale & Zhang, 2024), and thus inspires us to decompose it into two parts,

$$\tau U_E \dot{z}_E = \underbrace{[U_{ES}(z_S - z_E) + (1 - \alpha_L)U_{EF}(\mu_z - z_E)]}_{\text{Momentum } p, \text{ (Hamiltonian part)}} + \underbrace{[\alpha_L U_{EF}(\mu_z - z_E) + \sigma_z\sqrt{\tau U_E}\xi_t]}_{\text{Langevin part}},$$

where $\alpha_L \in [0, 1]$ denotes the proportion of Langevin sampling component. In this way, we can define the transformation matrix and rewrite,

$$\mathbf{z}_H \equiv \begin{pmatrix} z \\ p \end{pmatrix} = \underbrace{\begin{pmatrix} 1 & 0 \\ -[U_{ES} + (1 - \alpha_L)U_{EF}] & U_{ES} \end{pmatrix}}_{\mathbf{T}}\mathbf{z} + \underbrace{\begin{pmatrix} 0 \\ (1 - \alpha_L)U_{EF}\mu_z \end{pmatrix}}_{\mathbf{M}_2} \quad \text{(E13)}$$

We are interested in $\mathbf{z}_H$ dynamics, and investigate how the circuit parameters can be set to fulfill the Hamiltonian sampling. Without loss of generality, we consider a case of $\mu_z = 0$ that simplify the derivation of the $\mathbf{z}_H$ dynamics, which will make $\mathbf{M}_1 = \mathbf{M}_2 = 0$. And then,

$$\mathbf{z} = \mathbf{T}^{-1}\mathbf{z}_H \quad \text{where} \quad \mathbf{T}^{-1} = \frac{1}{U_{ES}}\begin{pmatrix} U_{ES} & 0 \\ U_{ES} + (1 - \alpha_L)U_{EF} & 1 \end{pmatrix}$$

Then we can derive the dynamics of $\mathbf{z}_H$,

$$\dot{\mathbf{z}}_H = \mathbf{T}\dot{\mathbf{z}}, \\ = \mathbf{T}[(\tau\mathbf{D_U})^{-1}\mathbf{F}_1\mathbf{z} + (\tau\mathbf{D_U})^{-1/2}\mathbf{\Sigma}_1\boldsymbol{\xi}_t], \\ = [\mathbf{T}(\tau\mathbf{D_U})^{-1}\mathbf{F}_1\mathbf{T}^{-1}]\cdot\mathbf{z}_H + \mathbf{T}(\tau\mathbf{D_U})^{-1/2}\mathbf{\Sigma}_1\boldsymbol{\xi}_t, \quad \text{(E14)}$$

where

$$\mathbf{T}(\tau \mathbf{D_U})^{-1}\mathbf{F}_1\mathbf{T}^{-1} = -\begin{pmatrix} \alpha_L U_{EF}(\tau U_E)^{-1} & -(\tau U_E)^{-1} \\ \beta_E & \beta_p \end{pmatrix},$$

$$\mathbf{T}(\tau \mathbf{D_U})^{-1/2}\mathbf{\Sigma}_1 = \begin{pmatrix} \sigma_z(\tau U_E)^{-1/2} & 0 \\ \sigma_p & 0 \end{pmatrix} \tag{E15}$$

and

$$\beta_E = -(\tau U_E)^{-1}[U_{ES} + (1-\alpha_L)U_{EF}]\alpha_L U_{EF} + (1-\alpha_L)(\tau U_S)^{-1}U_{SE}U_{EF},$$

$$\beta_p = (\tau U_E)^{-1}[U_{ES} + (1-\alpha_L)U_{EF}] + (\tau U_S)^{-1}U_{SE}, \tag{E16}$$

$$\sigma_p^2 = (\tau U_E)^{-1}[U_{ES} + (1-\alpha_L)U_{EF}]^2 \sigma_z^2$$

**Standard form of the Hamiltonian sampling dynamics**

We further convert the Eq. (E14) into the standard form of Hamiltonian sampling dynamics (Eq. 7), which corresponds to multiply the $z_E$ with the posterior precision $\Lambda$ and then compensate the $\Lambda^{-1}$ into the preceding matrix,

$$\begin{pmatrix} \dot{z}_E \\ \dot{p} \end{pmatrix} = -\begin{pmatrix} \alpha_L U_{EF}(\tau U_E)^{-1}\Lambda^{-1} & -(\tau U_E)^{-1} \\ \beta_E \Lambda^{-1} & \beta_p \end{pmatrix} \begin{pmatrix} \Lambda z_E \\ p \end{pmatrix} + \begin{pmatrix} \sigma_p & 0 \\ 0 & 0 \end{pmatrix} \boldsymbol{\xi}_t,$$

$$= -\begin{pmatrix} \alpha_L U_{EF}(\tau U_E)^{-1}\Lambda^{-1} & -\beta_E \Lambda^{-1} \\ \beta_E \Lambda^{-1} & \tau U_E \beta_p \beta_E \Lambda^{-1} \end{pmatrix} \begin{pmatrix} \Lambda z_E \\ (\tau U_E \beta_E)^{-1}\Lambda p \end{pmatrix} + \begin{pmatrix} \sigma_z(\tau U_E)^{-1/2} & 0 \\ \sigma_p & 0 \end{pmatrix} \boldsymbol{\xi}_t$$

The second equality comes from we have the freedom of determining the momentum $p$'s precision, and then we could choose a momentum precision to make sure the first matri on the RHS is anti-symmetric. Eventually, by using

$$U_{EF} = \lambda_z \Lambda, \quad \Lambda z_E = -\nabla_z \ln \pi(z_E), \quad \tau_X = \tau U_X \ (X = E, S),$$

We can convert the $(z_E, p)$ dynamics into the standard form of Hamiltonian sampling dynamics as shown in the main text (Eq. 16), i.e.,

$$\frac{d}{dt}\begin{bmatrix} z_E \\ p \end{bmatrix} = -\begin{bmatrix} \alpha_L \lambda_z(\tau U_E)^{-1} & -\beta_E \Lambda^{-1} \\ \beta_E \Lambda^{-1} & \tau_E \beta_p \beta_E \Lambda^{-1} \end{bmatrix} \begin{bmatrix} -\nabla_z \ln \pi(z_E) \\ (\tau_E \beta_E)^{-1}\Lambda \cdot p \end{bmatrix} + \begin{bmatrix} \sigma_z(\tau_E)^{-1/2} \\ \sigma_p \end{bmatrix} \boldsymbol{\xi}_t \tag{E17}$$

E.3   CONDITIONS FOR REALIZING HAMILTONIAN SAMPLING IN THE CIRCUIT

Realizing Hamiltonian sampling in the circuit requires we set the ratio between drift and diffusion terms appropriately in Eq. (E17).

$$\alpha_L \lambda_z \tau_E^{-1} = \sigma_z^2 \tau_E^{-1}/2 \tag{E18a}$$

$$\tau_E \beta_p \beta_E \Lambda^{-1} = \sigma_p^2/2 \tag{E18b}$$

Solving Eq. (E18a),

$$w_{EF} = \left(\frac{2}{\sqrt{3}}\right)^3 \mathsf{F}\alpha_L^{-1} \tag{E19}$$

Solving Eq. (E18b) by substituting Eq. (E16)

$$\Lambda^{-1}U_{EF}\big[ -(\tau U_E)^{-1}[U_{ES} + (1-\alpha_L)U_{EF}]\alpha_L + (1-\alpha_L)(\tau U_S)^{-1}U_{SE}\big]$$

$$\times \big[(\tau U_E)^{-1}[U_{ES} + (1-\alpha_L)U_{EF}] + (\tau U_S)^{-1}U_{SE}\big]$$

$$= \tau_E^{-2}[U_{ES} + (1-\alpha_L)U_{EF}]^2 \sigma_z^2/2.$$

To simplify notations, we define two intermediate variables about common factors in the above equation

$$h_E \equiv \tau_E^{-1}[U_{ES} + (1-\alpha_L)U_{EF}]; \quad h_S \equiv \tau_S^{-1}U_{SE}. \tag{E20}$$

And utilizing the Eq. (E18a), it simplifies the equation into

$$[-h_E \alpha_L + (1-\alpha_L)h_S](h_E + h_S) = \alpha_L h_E^2$$

Reorganizing the above equation into a quadratic equation of $h_E$,

$$2\alpha_L \cdot h_E^2 + (2\alpha_L - 1) \cdot h_S h_E + (\alpha_L - 1)h_S^2 = 0,$$

Then the root of the $h_E$ is

$$h_E = h_S \frac{(1 - 2\alpha_L) \pm \sqrt{1 + 4\alpha_L - 4\alpha_L^2}}{4\alpha_L} \equiv Q(\alpha_L) \cdot h_S \qquad \text{(E21)}$$

Combining the expression of $h_E$ in Eq. (E20),

$$\tau_E^{-1}[U_{ES} + (1 - \alpha_L)U_{EF}] = Q(\alpha_L) \cdot \tau_S^{-1} U_{SE}$$

Then substituting the detailed expression of $U_{EF}$, $U_{SE}$, $\tau_E$, and $\tau_S$ into the above equation, we have

$$\left(U_E^{-1} R_S\right) \cdot w_{ES} - \left[(1 - \alpha_L)U_E^{-1} R_F\right] \cdot w_{EF} = \left[Q(\alpha_L)U_S^{-1} R_E\right] \cdot w_{SE}, \qquad \text{(E22)}$$

which is the Eq. (17) in the main text.

### E.4 NATURAL GRADIENT HAMILTONIAN: DETERMINING THE MOMENTUM PRECISION IN THE CIRCUIT

Eq. (E17) suggests the momentum precision in the circuit dynamics is

$$\Lambda_p \equiv (\tau_E \beta_E)^{-1} \Lambda,$$

which should be proportional to the inverse of the Fisher inforamtion of the stimulus, $G(z)$ (Eq. 7). We next verify whether this can be satisfied in the circuit dynamics.

Substituting the expression of $\beta_E$ in Eq. (E16) into the above equation and using the simplified notation $h_E$ (Eq. E20), we have

$$\Lambda_p = \tau_E^{-1}\Lambda\left([-\alpha_L h_E + (1 - \alpha_L)h_S]U_{EF}\right)^{-1}$$

Utilizing the relation between $h_E$ and $h_S$ in Eq. (E21),

$$\Lambda_p = \tau_E^{-1}\Lambda\left([-\alpha_L Q(\alpha_L) + (1 - \alpha_L)]h_S U_{EF}\right)^{-1},$$

$$= \underbrace{\frac{1}{[-\alpha_L Q(\alpha_L) + (1 - \alpha_L)]}}_{\approx \text{ const.}} \underbrace{\frac{\Lambda}{U_{EF}}}_{\lambda_z^{-1}} \frac{1}{\tau_E h_S}.$$

Here the first term of $\alpha_L$ about the proportion of Langevin sampling can be treated as a constant, and the $\lambda_z$ is also a constant that doesn't change with the network activity. Substituting the detailed expression of $\tau_E$ and $h_S$ (Eq. E20)

$$\Lambda_p \propto (\tau_E h_s)^{-1} = \frac{U_S}{U_E U_{SE}} = U_E^{-1},$$

where the last equality comes from $U_S = U_{SE}$ in the equilibrum state (Eq. (E7)) Furthermore, from the bump height dynamics in the augmented circuit with SOM (Eq. E6), and using similar analysis in Eq. (C22)

$$U_E = (U_{EE} + U_{ES}) + U_{EF},$$
$$= (U_{EE} + U_{ES}) + \lambda_z G(z),$$

which clearly shows the $U_E$ in the augmented circuit increases with the Fisher information of the stimulus $G(z)$. Since the momentum precision $\Lambda_p$ is inversely proportional to $U_E$, it decreases with the stimulus Fisher information $G(z)$, which is consistent with the natural gradient Hamiltonian sampling (Eq. 7).

## F CIRCUIT SIMULATION PARAMETERS AND DETAILS

### F.1 CRITICAL WEIGHT

To scale the connection strengths in our network model, we use a critical recurrent connection strength as a reference point. This critical strength is defined as the smallest value that allows the network to maintain persistent activity even when there is no feedforward input.

Table 1: PARAMETERS FOR HAMILTONIAN SAMPLING

| PARAMETER | VARIABLE | VALUE |
|---|---|---|
| E time constant | $\tau$ | 1 |
| Connection width | $a_E$ | 40° |
| Num. of E neurons | $N_E$ | 180 |
| Fano factor | F | 0.5 |
| Normalization | $w_{EP}$ | $5 \times 10^{-4}$ |
| Feedforward weight | $w_{mm}^{EF}$ | $0.2\sqrt{2}w_c$ |
| Coupling Weight | $w_{mn}^{EE}$ | $0.8w_c$ |

In the absence of feedforward input, the stationary state of circuit's bump height satisfies (Eqs. E6 - E7),

$$U_E = \frac{\rho}{\sqrt{2}} R_E \left[ w_{EE} + \frac{\rho}{\sqrt{2}} w_{ES} g_S w_{SE} \right],$$

$$U_S = \frac{\rho}{\sqrt{2}} \frac{a_E}{a_S} w_{SE} R_E. \tag{F1}$$

Furthermore, the firing rate of the E population, $R_E$, is related to its input $U_E$ by the activation function defined in Eq. (C5). Substituting this expression for $R_E$ into Eq. (F1) allows us to write an equation solely in terms of $U_E$:

$$U_E = \frac{\rho U_E^2}{\sqrt{2} + 2\sqrt{\pi} k \rho a_E U_E^2} \left[ w_{EE} + \frac{\rho}{\sqrt{2}} w_{ES} w_{SE} g_S \right].$$

Assuming $U_E \neq 0$ (for persistent activity), we can divide by $U_E$ and rearrange the equation into a quadratic form for $U_E$:

$$2\sqrt{\pi} k \rho a_E U_E^2 - \rho \left[ w_{EE} + \frac{\rho}{\sqrt{2}} w_{ES} w_{SE} g_S \right] U_E + \sqrt{2} = 0.$$

Let $w_c = w_{EE} + \frac{\rho}{\sqrt{2}} w_{ES} w_{SE} g_S$. This quadratic equation for $U_E$ has real solutions if and only if its discriminant is non-negative ($\rho^2 w_c^2 - 8\sqrt{2\pi} k \rho a_E \geq 0$). The smallest value of $w_c$ that permits non-zero persistent activity occurs when the discriminant is zero, i.e.,

$$w_c^2 = \frac{8\sqrt{2\pi} k a_E}{\rho}. \tag{F2}$$

The network parameters used in our simulations are provided in Table 1. This includes parameters like the number of neurons ($N_E = 180$, $N_S = 180$) distributed over a feature space of width $w_z = 360°$, leading to a neuronal density $\rho = N/w_z$. So the critical weight value is calculated as:

$$w_c = 2\sqrt{2}(2\pi)^{1/4} \sqrt{ka/\rho} \approx 0.896. \tag{F3}$$

The intensity of the feedforward input is then scaled relative to $U_c$, which is the peak synaptic input to the E population that is self-sustained by the E recurrent connections at their critical strength $w_c$, in the absence of feedforward input and SOM inhibition. $U_c$ is given by:

$$U_c = \frac{w_c}{2\sqrt{\pi} ka}. \tag{F4}$$

F.2    PARAMETERS FOR NETWORK SIMULATION

For the reduced network with only PV and excitatory neuron, the network parameters is set as following. This parameter set applies for a single circuit sampling a 1D stimulus posterior, and coupled circuits sampling multivariate stimulus posteriors. For 1D and 2D, the parameters are the same aside there are not couping weight for 1d case.

For the equilibrium state analysis depicted in Figure 2, the network is first initialized using an input intensity identical to that of subsequent simulation phases, in order to remove the influence of

Table 2: Parameters for network

| PARAMETER | VARIABLE | VALUE |
|---|---|---|
| Number of trials | | 500 |
| Simulation time | $T$ | 500.0 |
| Time step | $dt$ | 0.01 |
| Recording start | $t_{steady}$ | 50 |
| Input position | $\mu$ | 0 |
| Initial mean eq | $\mu_0$ | 0 |
| Initial var eq | $V_0$ | 30 |

Table 3: PARAMETERS FOR HAMILTONIAN SAMPLING

| PARAMETER | VARIABLE | VALUE |
|---|---|---|
| Num. of SOM | $N_S$ | 180 |
| SOM time constant | $\tau_S$ | 1.0 |
| SOM connection width | $a_S$ | 37.4° |
| E to SOM connection width | $a_{SE}$ | 34.6° |
| SOM to E connection width | $a_{ES}$ | 20° |
| SOM to E connection weight | $w_{ES}$ | $0.6w_c$ |

non-equilibrium bump height. During this initialization, the input position varies across trials, drawn from a Gaussian distribution with mean $\mu_0$ and variance $V_0$.

After allowing the network's bump height to reach equilibrium post-initialization, the input position is then set to match the mean of the network's activity bump. The simulation proceeds for a duration of $50\tau$, using an integration time step of $0.01$ time units. The first 20 time steps of this period are discarded to avoid transient effects. Following this, the input position is fixed at 0, and the network is simulated for an additional $450\tau$ with the same integration step.

Throughout the latter $450\tau$ simulation, the bump position is recorded to calculate the KL divergence between the network's evolving state and a target posterior distribution. The network state at the end of the initialization phase serves as the reference for the initial KL divergence value.

For comparison, a separate Langevin sampling process is performed. This sampling is initialized using the network's bump position from the end of its initialization phase. The Langevin sampling then runs for a duration of $450\tau$, also using an integration time step of $0.01$ time units.

For the non-equilibrium state depicted in Figure 2, the network is initially prepared by applying a substantially smaller input signal, denoted as $scale_{ini}$. This input is administered uniformly to all neurons for a duration of $20\tau$ to initialize the network. After this initialization phase, the input to each neuron is then adjusted to its designated operational value.

For the natural gradient (NG) sampling procedure, the starting position is set to the 'bump' location observed at the final step of the Continuous Attractor Neural Network (CANN) model's initialization.

For the different recurrent weight, we fix input intensity $R_F = 3$. We get time constant by getting the cross-corelation of bump postion simulated from the network and fit the exponential function to get the time constant.

Hamiltonian sampling parameters mostly mirror the previous set, but differ by including connection parameters that define interactions between SOM and excitatory neurons. For 500 trials and simulation $500\tau$, it takes 2 hours on the 512GB cpu hpc.

### F.2.1 NUMERICAL ESTIMATE OF THE STIMULUS PRIOR IN COUPLED CIRCUITS

We numerically estimate the subjective bivariate stimulus prior stored in the coupled circuits. Given a combination of circuit parameters, we ran a large ensemble of stochastic network simulations. From

the spatio-temporal firing rate patterns in each circuit $\mathbf{r}_m$, we decoded instantaneous population vectors $z_m$ in each time bin in each trial. Then we concatenate the $z_m$ from two circuits together, $\mathbf{z} = (z_1, z_2)$, and estimate its mean $\boldsymbol{\mu_z}$ and covariance $\boldsymbol{\Sigma_z}$, which are used to parameterize the Gaussian sampling distribution, i.e., $p(\mathbf{z}) = \mathcal{N}(\boldsymbol{\mu}_z, \boldsymbol{\Sigma}_z)$.

And then we search the prior precision matrix $\mathbf{L}$ under which the posterior is closet to the sampling distribution $p(\mathbf{z})$,

$$\hat{\mathbf{L}} = \arg\min_{\mathbf{L}} D_{KL}\left[\pi(\mathbf{z}) \| p(\mathbf{z})\right]$$

where the posterior $\pi(\mathbf{z})$ is calculated based on the parameter $\mathbf{L}$ to be estimated,

$$\pi(\mathbf{z}) = \mathcal{N}\left(\boldsymbol{\mu_z}, \boldsymbol{\Omega}^{-1}\right), \quad \text{with} \quad \boldsymbol{\Omega} = \boldsymbol{\Lambda} + \mathbf{L}, \quad \boldsymbol{\mu_z} = \boldsymbol{\Omega}^{-1}\boldsymbol{\Lambda}\boldsymbol{\mu},$$

and the likelihood mean $\boldsymbol{\mu}$ and precision $\boldsymbol{\Lambda}$ are directly estimated from the received feedforward inputs (Eq. C18),

$$\mu_{z,m} = \frac{\sum_j \mathbf{r}_m(\theta_j)\theta_j}{\sum_j \mathbf{r}_m(\theta_j)}, \quad \Lambda_m = a^{-2}\sum_j \mathbf{r}_m(\theta_j) \approx \sqrt{2\pi}\rho a^{-1}R_m^F. \tag{F5}$$

### F.2.2 THE VECTOR FIELD (DRIFT TERM) OF CIRCUITS' SAMPLING DYNAMICS

For both the diagonal-Fisher natural-gradient Langevin sampler and the full-Fisher method, we can directly compute the gradient at each point in parameter space, evaluate the Fisher information (either the full matrix or just its diagonal), and then derive the corresponding vector field from this information.

In the case of our CANN (Continuous Attractor Neural Network) model, constructing the equilibrium vector field requires a slightly different approach. The goal is to observe how the position of the bump (i.e., the localized peak of neural activity) shifts in response to changes in the input. To do this, we first stabilize the bump at a reference location. Specifically, we apply a fixed external input centered at $(x_0, y_0)$ and run the CANN dynamics until the bump height reaches equilibrium. In our experiments, this equilibration phase lasted for 20 time constants $(20\tau)$.

Once the bump has stabilized, we perturb the input by shifting it to a new position $(x_1, y_1)$, and observe how the bump position responds. The resulting displacement of the bump provides the vector at the new point, essentially showing how the internal state of the network changes in response to this small input shift. Analytically, this shift can be expressed as moving the input from $(x_0, y_0)$ to $(x_2, y_2) = (x_0, y_0) + \Lambda^{-1}\Omega(x_1 - x_0, y_1 - y_0)$, where $\Lambda^{-1}\Omega$ captures the relationship between input space and the internal dynamics of the bump.

Because our 2D network structure implicitly encodes a prior, shifting the bump corresponds to translating the mean of the posterior distribution. Repetition of this process across a grid of input locations $(x_2, y_2)$, we can scan the whole bump position grid and then we can systematically map out the equilibrium vector field of the CANN. This field describes how the network's internal estimate-the bump position-evolves in response to perturbations in the input.

### F.3 PARAMETERS FITTING

In our attractor network, the bump position $z(t) = (z_E(t), z_S(t))^\top$ is determined by the connection between Excitoory and SOM populations. The dynamics are described by equations(E10).

By introducing a compact notation and collecting terms into matrix-vector form, we specifically define the state as $\mathbf{z} = (z_E, z_S)^\top$ and the 2D dynamics as:

$$\dot{\mathbf{z}} = \mathbf{D_U^{-1}}\mathbf{F}_1\mathbf{z} + \mathbf{D_u^{-1}}\mathbf{M}_1 + \boldsymbol{\Sigma}_1\xi_t \tag{F6}$$

where $\boldsymbol{D}_U^{-1}\boldsymbol{F} \in \mathbb{R}^2$ which is the drift matrix, $M_1 \in \mathbb{R}^2$ is a constant input, and $\Sigma\xi_t$ is noise term.

Convert into the form of transition probability:

$$\mathbf{z}_{t+\Delta t} \sim \mathcal{N}\left((\mathbf{I} + \mathbf{D_u^{-1}}\mathbf{F}_1\Delta t)\mathbf{z}_t + \mathbf{D_u^{-1}}\mathbf{M}_1\Delta t, \mathbf{Q}\right), \tag{F7}$$

where

$$\mathbf{D_u}^{-1}\mathbf{F}_1 = \begin{bmatrix} h_{ES} + h_{EF} & -h_{ES} \\ h_{SE} & -h_{SE} \end{bmatrix}, \mathbf{Q} = \mathbf{\Sigma_1}\Delta t \tag{F8}$$

and $\quad h_{ES} = -U_E^{-1}U_{ES}, \quad h_{EF} = -U_E^{-1}U_{EF}, \quad h_{SE} = -U_S^{-1}U_{SE}$ are time-rescaled synaptic coefficients.

Because the noise enters the network only through the excitatory population. We therefore estimate the four unknown parameters $\{h_{SE}, h_{ES}, h_{EF}, \sigma_z\}$ in two consecutive steps.

From the noiseless second equation, we have $\dot{z}_S = U_{SE}(z_E - z_S)$ with the closed-form discrete update $z_S(t+\Delta t) - z_S(t) = U_{SE}[z_E(t) - z_S(t)]\Delta t$. Averaging over a trajectory of length $T$ gives an unbiased estimator

$$\widehat{U}_{SE} = \frac{\langle z_S(t+\Delta t) - z_S(t)\rangle_t}{\Delta t \langle z_E(t) - z_S(t)\rangle_t}, \tag{F9}$$

so no optimization is required.

Conditioned on $z_S$, the excitatory coordinate follows a scalar Ornstein–Uhlenbeck process

$$z_E(t+\Delta t) = z_E(t) + \Delta t\big[(h_{ES} + h_{EF})z_E(t) - h_{SE}z_S(t)\big] + \sigma_z\sqrt{\Delta t}\,\xi_t. \tag{F10}$$

Then we used maximum likelihood estimation (MLE) to estimate the parameters of $\{h_{SE}, h_{ES}, h_{EF}, \sigma_z\}$ in the above equation. All parameters are regressed on a data segment of 1000 samples, corresponding to $10\,\tau$. The parameters are explicitly reparameterized in terms of their biological interpretation and optimized via stochastic gradient descent (Adam), enabling stable and interpretable system identification.

After obtaining the MLE estimate of $\{h_{SE}, h_{ES}, h_{EF}, \sigma_z\}$ for each data set, we numerically find the transformation $\mathbf{T}$ matrix (Eq. E13) by directly estimating the values of $U_E$, $U_S$ and $\alpha_L$ from network activities. Eventually, we use the estimated $\mathbf{T}$ matrix to convert the $(z_E, z_S)$ into $(z_E, p)$ as described in Eq. (E13).

We evaluated $\Lambda_p$ under 11 values of feedforward input intensity, $R_F \in \{11, 13, \ldots, 23\}$, and found the decreasing tread of kinetic energy. The decreasing trend confirms the theoretical prediction that a stronger external drive reduces the effective momentum budget required for accurate sampling. All experiments were performed on a compute node equipped with 40 CPU cores and one NVIDIA A100 GPU; the full pipeline completed in about 26 hours.

### F.4 Embedding canonical circuits as a latent space sampler in VAE framework

**Dataset and augmentation**. As a proof-of-concept example, we consider the MNIST handwritten digit dataset. To mimic a simple physical stimulus space, each grayscale image was augmented by a *rotation* and a *contrast rescaling*. Let $\mathbf{x}_0$ denote the original $28 \times 28$ image (pixel values in $[0,1]$). For each training sample we drew

$$z \sim \mathcal{U}[-z_{\max}, z_{\max}], \qquad c \sim \mathcal{U}[c_{\min}, c_{\max}], \tag{F11}$$

with $z_{\max} = 180°$, $c_{\min} = 0.1$, $c_{\max} = 1.0$. We then applied a rotation by angle $\theta$ followed by a contrast adjustment with factor $c$ using bilinear interpolation and zero padding.

**Model architecture**. We connect the encoder, canonical circuit, and decoder in serial (Fig. 5A). Given an input image $\mathbf{x}$, we pass it into the encoder that is a multi-layer perception (MLP) and is supposed to output normalized latent variables:

$$(\hat{z}, \hat{c}) = f_{Encoder}(\mathbf{x}), \tag{F12}$$

where $\hat{z}$ and $\hat{c}$ represent the estimated orientation and the contrast, respectively. Then we pass them into Eq. (1e) to generate a population feedforward input $\mathbf{r}_F$ to canonical circuit,

$$\mathbf{r}_F(\theta) = \hat{c}R_F \exp[-(\theta - \hat{z})^2/2a^2]. \tag{F13}$$

Next, the feedforward input $\mathbf{r}_F$ is fed into the canonical circuit dynamics (Eqs. 1a-1b) that produces the bump population firing rate $\mathbf{r}_E(\theta, t)$ over time. After the circuit dynamics enters the equilibrium, we read out the bump response location $z_E$ and height $R_E$ from the $\mathbf{r}_E(\theta, t)$, where $z_E$ is regarded

as a sample drawn from the orientation likelihood. Then $(z_E, R_E)$ forms the input to the decoder that is supposed to output a reconstructed image,

$$\hat{\mathbf{x}} = f_{Decoder}(z_E, R_E) \tag{F14}$$

In our example, both encoder and decoder are MLPs with two hidden layers and ReLU activation function.

**Training the encoder and decoder**. Since the latent sampler is given in VAE and it will be implemented by our canonical recurrent circuit, we don't train the recurrent circuit while only train the encoder and decoder. We trained the encoder in a purely supervised fashion by minimizing the mean-squared error between predicted latent variable, $(\hat{z}, \hat{c})$, and the target latent $(z, c)$. Similarly, the decoder also undergoes supervised training that reconstructs the augmented images $\hat{\mathbf{x}}$ given the true latent parameters $(\hat{z}, \hat{c})$, minimizing the pixel-wise Bernoulli negative log-likelihood. During training, we optionally replaced the ground-truth $(z, c)$ by the well-trained encoder predictions $(\hat{z}, \hat{c})$, effectively training a full autoencoder.

All models were implemented in PyTorch. For both the encoder and decoder, we used ReLU nonlinearities and trained with Adam optimizer (learning rate $10^{-3}$, default momentum parameters). The encoder was trained for 100 epochs with batch size 128, and the decoder for 200 epochs with batch size 256.

## G GENERALIZATION

### G.1 NON-FLAT PRIOR

If we want to store a non-uniform marginal prior needs to break the symmetry of E neurons, i.e., the tuning curves of all E neurons (Fig. A1C) cannot be homogeneous (aligned after translation in stimulus subspace) and uniformly cover the stimulus subspace. Previous studies suggested we can introduce the heterogeneity of tuning height, width and tuning's distribution on the stimulus subspace to store non-uniform marginal priors. Regarding the circuit mechanism, the tuning heterogeneity for non-uniform marginal prior may come from 1) an external prior input that may from higher cortex, or 2) internally stored in the recurrent weights in the network model which can be realized by introducing an extra heterogeneous recurrent weight component superimposed on the translation-invariant recurrent weight matrix. We provide brief analysis of the possibility 1) by showing an external prior input can still maintain the relation between bump height and the FI. For simplicity, we consider the external prior input is constant over time and it also has a Gaussian profile that is added into the circuit dynamics (Eq.1a),

$$\rho\textstyle\sum_X (\mathbf{W}_{EP} * \mathbf{r}_P)(\theta, t), \quad \mathbf{r}_P(\theta, t) = R_P \exp[-(\theta - z_P)^2 / 2a^2]$$

It will become,

$$\tau\dot{\mathbf{u}}_E(\theta, t) = -\mathbf{u}_E(\theta, t) + \rho\textstyle\sum_X (\mathbf{W}_{EX} * \mathbf{r}_X)(\theta, t) + \rho\textstyle\sum_X (\mathbf{W}_{EP} * \mathbf{r}_P)(\theta, t) + \sqrt{\tau \mathsf{F}[\mathbf{u}_E(\theta, t)]_+} \xi(\theta, t).$$

Mathematically, the external prior input is similar to the feedforward input. By performing the same math calculations, the Eqs. 3a and 3b are updated into with a new term,

$$\text{Position}: \ \dot{z}_E \approx (\tau U_E)^{-1} U_{EF}(\mu_z - z_E) + (\tau U_E)^{-1} U_{EP}(\mu_p - z_E) + \sigma_z (\tau U_E)^{-1/2}\xi_t \tag{G1a}$$

$$\text{Height}: \ \dot{U}_E \approx \tau^{-1}[-U_E + U_{EE} + U_{EF} + U_{EP}] + \sigma_U (\tau^{-1} U_E)^{1/2}\xi_t, \tag{G1b}$$

where the $U_{EP}$ and $\mu_p$ characterizes the precision and mean of the non-uniform marginal prior. Since the drift term of $z_E$ dynamics contains feedforward input (1st RHS term) and the prior input (2nd RHS term), circuit can sample the posterior with non-uniform marginal prior. When z reaches equilibirum, the distribution will be the posterior,

$$p(z) \propto exp[\sqrt{2\pi}\rho R_f a^{-1}(\mu_z - z_E)^2 + \sqrt{2\pi}\rho R_p a^{-1}(\mu_p - z_E)^2]$$

In addition, the updated $U_E$ dynamics also receives the $U_{EP}$ representing the prior precision, suggesting the bump height $U_E$ still encodes the updated posterior FI. Because the log-posterior and its Fisher information are additive ($G_{posterior} = G_{likelihood} + G_{prior}$), the total bump height $U_E$ naturally becomes the sum of the input heights (e.g., $U_{EF} + U_{EP} \propto FI$), with each height representing its component of the total information.

## G.2 VON MISES CASE

Continuing the above modification of the recurrent circuit, we can further investigate how the circuit can implement NG sampling of 1D von Mises likelihoods (the circuit still stores a uniform prior), e.g., $p(\mu_z|z) \propto \exp[\kappa \cos(z - \mu_z)]$.

The Fisher information (FI) under the von Mises posterior is

$$G(z) = \mathbb{E}_{p(\mu_z|z)}[-\nabla_z^2 \ln p(\mu_z|z)] = \int [-\nabla_z^2 \ln p(\mu_z|z)] p(\mu_z|z) d\mu_z$$

where the negative Hessian of the log-likelihood is

$$-\nabla_z^2 \ln p(\mu_z|z) = \kappa \cos(z - \mu_z)$$

and thus the FI is

$$G(z) = \int \kappa \cos(z - \mu_z) \frac{\exp[\kappa \cos(z - \mu_z)]}{2\pi I_0(\kappa)} d\mu_z = \kappa \frac{I_1(\kappa)}{I_0(\kappa)}$$

where $I_0(\kappa)$ and $I_1(\kappa)$ are modified Bessel function of the first kind with zeroth and first order respectively.

Although we see the FI of the von Mises likelihood doesn't depend on $z$ , i.e., the local geometry. Our preliminary analysis find the recurrent circuit dynamics in response to an observed feature $\mu_z$ within a trial is not able to evaluate the expectation over $\mu_z$. Instead, we find within a trial, the circuit corresponds to approximating the observed FI by using the mean of the likelihood, i.e.,

$$G(z) = \mathbb{E}_{p(\mu_z|z)}[\kappa \cos(z - \mu_z)] \approx \kappa \cos(z - x)\big|_{x=\mu_z} = \kappa \cos(z - \mu_z)$$

Therefore, within a trial the approximated FI is dependent on local geometry. Once we follow the same theoretical protocols and update the bump height $U_E$ dynamics in the case of von Mises kernel, we find the $U_E$ is a function of $\kappa \cos(z - \mu_z)$ shown as the last term of the following equation, i.e.,

$$U_E = \frac{\rho w_{EE}}{e^{a/2}} \frac{I_0(a)}{I_0(a/2)^2} R + \frac{8 + a}{8 I_0(a)} R_F + \frac{a \cos(\mu_z - z)}{8 I_0(a)} R_F$$

