# OpenReview forum: "Natural gradient Bayesian sampling automatically emerges in canonical cortical circuits"
_ICLR.cc/2026/Conference — Submitted to ICLR 2026_

### Official Review · Reviewer_uXx1 · 2025-10-25

**Soundness:** 4
**Presentation:** 1
**Contribution:** 3
**Rating:** 4
**Confidence:** 3

**Summary:**

This paper presents a theoretical analysis of how canonical cortical circuits, composed of excitatory (E) and inhibitory (PV, SOM) neurons, can implement natural gradient Bayesian sampling. The core contribution is the finding that, under a uniform prior, the bump height of the E-neuron population response encodes the Fisher Information (FI) of the posterior. This mechanism dynamically controls the sampling time constant of the bump's position, allowing the circuit to biologically implement Natural Gradient Langevin Sampling (NGLS). The authors further demonstrate that incorporating SOM neurons, which introduce a momentum term, upgrades the circuit to perform Natural Gradient Hamiltonian Sampling (NGHS). The work also connects other computational strategies, such as annealing and regularization, to the emergent properties of the E-I circuit dynamics.

**Strengths:**

1. The paper provides a comprehensive theoretical bridge between the dynamics of canonical E-I circuits and advanced Bayesian sampling algorithms.

2. It demonstrates how distinct neuron populations and their specific interactions can mechanistically implement components of these advanced sampling strategies.

3. It also offers novel insights into the computational role of non-equilibrium dynamics as an innate annealing strategy that can accelerate sampling.

4. The theoretical claims are well-supported by numerical simulations.

**Weaknesses:**

A major concern is that the work's core findings appear to be an incremental step over Sale & Zhang (2024). The authors acknowledge their work builds on this previous study , which had already established the circuit's Langevin and Hamiltonian sampling properties. The main new finding is that the E-neuron response height encodes the FI, thus upgrading the circuit to NGLS/NGHS. However, this critical finding is true only when the prior is uniform. This assumption limits the generality and remains a constraint on the work's overall impact.

The writing and clarity of the paper can be further improved. Some notations are a little bit confusing or not adequately explained in the main text. (E.g., $\xi$ in Eq. (1a), $p(s)$ in "We will leave the subjective prior p(s)...", and some inconsistent notations between main texts and appendix.) This makes it difficult to fully understand the theoretical analysis. Besides, although the figures are generally neat and beautiful, legends in Figure 2E&F&K are confusing. Different colors used in these figures are not explained.

**Questions:**

See the weakness section for my major concerns.

1. Could the authors please elaborate on the significance of their findings beyond Sale & Zhang (2024)? Given that the key NGLS/NGHS finding is constrained to a uniform prior, how much of the circuit's behavior is already captured by LS/HS models from the previous work?

2. How would the proposed mechanism (E bump height encodes FI) break down if a non-uniform prior is introduced? Would the circuit still perform some approximation of NGLS, or would the direct link between bump height and FI be lost?

---

> ### Author Response · Authors · 2025-11-21
>
> Thanks for the reviewer’s positive comments about the theoretical discovery of the NG sampling in canonical cortical circuits. We will also proofread the manuscript to improve the writing. Below, we reply to your weaknesses and questions one by one.
>
> > Could the authors please elaborate on the significance of their findings beyond Sale & Zhang (2024)? Given that the key NGLS/NGHS finding is constrained to a uniform prior, how much of the circuit's behavior is already captured by LS/HS models from the previous work?
>
> Please refer to the section “The novelty of the present paper compared with Sale 2024” in the global official comments.
>
> > How would the proposed mechanism (E bump height encodes FI) break down if a non-uniform prior is introduced? Would the circuit still perform some approximation of NGLS, or would the direct link between bump height and FI be lost?
>
> A short answer is the relation between E bump height with FI can still hold with a non-uniform prior. Below we provide detailed analysis. We also augment the generalization in Discussion in the revised manuscript.
> Storing a non-uniform prior is an important extension of our circuit model. The recurrent inputs between two coupled circuits (Sec. 5) store a non-uniform associative prior about the correlation between two latent stimuli (Eq. 14). Nevertheless, each of the marginal distribution of the 2D associative prior in the coupled circuits is still uniform, because the matrix $L$ is degenerate with zero determinant (Eq. 13). This comes from the translation-invariant recurrent kernel (Eq. 1d; and convolution $*$ in Eq. 1a).
> The above analysis suggests storing a __non-uniform marginal prior__ needs to __break the symmetry__ of E neurons, i.e., the tuning curves of all E neurons (Appendix Fig. A1C) cannot be homogeneous (aligned after translation in stimulus subspace) and uniformly cover the stimulus subspace. Previous studies suggested we can introduce the heterogeneity of tuning height, width and tuning’s distribution on the stimulus subspace to store non-uniform marginal priors (Ganguli, NIPS 2010; Neural Computation 2014). This was listed in our future study as listed in line 473-476 in our original manuscript.
> Regarding the circuit mechanism, the tuning heterogeneity for non-uniform marginal prior may come from 1) an external prior input that may from higher cortex, or 2) internally stored in the recurrent weights in the network model which can be realized by introducing an extra heterogeneous recurrent weight component superimposed on the translation-invariant recurrent weight matrix. As a proof-of-concept example, we provide brief analysis of the possibility 1) by showing _an external prior input can still maintain the relation between bump height and the FI_. For simplicity, we consider the external prior input is constant over time and it also has a Gaussian profile that is added into the circuit dynamics (Eq.1a),
>
> $$
> \rho {\textstyle \sum_{X}} (\mathbf{W}_{EP}* \mathbf{r}_P)(\theta,t) , \quad \mathbf{r}_P(\theta,t) = R_P \exp[ -(\theta - z_P)^2 / 2a^2]
> $$
>
> Mathematically, the external prior input is similar to the feedforward input. By performing the same math calculations, the Eqs. 3a and 3b are updated into with a new term,
>
> Position:
> $$
>   \frac{d z_E}{dt}  \approx (\tau U_E)^{-1} U_{EF} (\mu_z - z_E)+(\tau U_E)^{-1} U_{EP} (\mu_p - z_E) + \sigma_z (\tau U_E)^{-1/2} \xi_t
> $$
>
> Height
> $$
>   \frac{d U_E}{dt} \approx \tau^{-1} [- U_E + U_{EE} + U_{EF} + U_{EP}] + \sigma_U (\tau^{-1} U_E)^{1/2} \xi_t,
> $$
>
> where the $U_{EP}$ and $\mu_p$ characterizes the precision and mean of the non-uniform marginal prior.
> Since the drift term of $z_E$ dynamics contains feedforward input (1st RHS term) and the prior input (2nd RHS term), circuit can sample the posterior with non-uniform marginal prior. In addition, the updated $U_E$ dynamics also receives the $U_{EP}$ representing the prior precision, suggesting the bump height $U_E$ still encodes the updated posterior FI.
> Furthermore, this framework is not limited to Gaussian distributions. Please refer to our global official comment and our new Fig. ?? for detailed information.
>
>
> ### Legends
>
> The colormap of Fig.2E are same as Fig.2C, Fig.2J&K has the same color map with Fig.2F. So the solid line in Fig 2 K are the network results with or without recurrent weight. And we apologize for the confusing legend and we will update it in the revised manuscript soon.

---

> > ### Comment · Reviewer_uXx1 · 2025-11-27
> >
> > Thank you for the response, which addressed some of my concerns. However, the overall presentation of the work can still be improved. In addition, after reviewing the comments by reviewer BBUn, I also agree that this work has some overlap with the submission https://openreview.net/forum?id=BpBW4gJofo, which is not trivial. So I will keep my initial rating.

---

> > > ### Author Response · Authors · 2025-11-27
> > >
> > > Thanks very much for your response. We will improve our presentation in the revised manuscript by taking reviewers’ comments. If you still have some concerns, feel free to bring up and we are happy to resolve them during the discussion period.
> > >
> > > Furthermore, regarding to the overlap with the paper mentioned by reviewer BBUn, we provided a detailed distinction in the section 'The relationship of this manuscript to the other paper' within our initial reply to reviewer BBUn. We copy that reply in the text below for your reference. PS: we hope our reply has successfully addressed reviewer BBUn’s concern of the overlap since his further reply doesn’t mention that issue.
> > >
> > > > The relationship of this manuscript to the other paper
> > >
> > > We have checked the paper mentioned by the reviewer. And we believe the two papers has substantial differences and thus our paper doesn’t violate the research integrity issues (e.g., plagiarism, dual submission)  policy and doesn’t have ethical issue.
> > > Although both papers study the same circuit model, and the other paper also briefly mentions the adaptive sampling size, the two papers study different computational tasks and have different focuses. Our paper comprehensively studies on how the network realizes NG sampling of __static__ univariate and multivariate posteriors, and identifies the computational approximations in NG sampling in the network. By contrast, the other paper focuses on the sampling of __dynamic__ univariate posterior that changes over time, without presenting the multivariate posterior cases. And the adaptive step size in the other paper is more like a side note in the introduction of algorithmic section, without comprehensive results in the Result section (only briefly mentioned in original manuscript lines 367-372). Therefore, we think the two papers have substantial differences.
> > > Technically, the other paper only projected the dynamics onto the position mode, while the present study projected the circuit dynamics onto both position mode (Eq. 12a) and the __height__ mode (Eq. 12b). The height mode projection is the cornerstone for identifying the NG sampling in the circuit dynamics. These have yielded many novel results that are absent in the other paper:
> > > 1. The height $U_E$ (Eq. 3b) acts as the **dynamic time constant** of the position dynamics (Eq. 3a).
> > > 2. It dynamically **encodes the Fisher Information** of the probability distribution.
> > > 3. We analytically identify an embedded the __annealing strategy__ in the circuit (Eq. 3b and Sec. 4.3).
> > > 4. We identify the NG Hamiltonian sampling in the circuit by proving the precision of momentum decreases with FI, which is absent in the other paper.
> > > Combined, we believe there is no research integrity issue of the two papers, based on their significant differences of the computational tasks and conclusions.

---

### Official Review · Reviewer_BBUn · 2025-10-28

**Soundness:** 2
**Presentation:** 2
**Contribution:** 2
**Rating:** 2
**Confidence:** 4

**Summary:**

There has in recent years been much interest in sampling-based probabilistic inference in neural networks as a model for various probabilistic computations in the brain. This paper builds on a line of work by Wen-Hao Zhang and collaborators, in particular [Sale and Zhang, NeurIPS 2024](https://proceedings.neurips.cc/paper_files/paper/2024/file/f463d31ed2fdd7b0ec585c041ec1baa8-Paper-Conference.pdf), that aims to show that the dynamics of the location of a bump attractor state in a mean-field-style model for the canonical cortical microcircuit can sample interesting distributions. Its contribution is to try to show that the dynamics of the bump can approximate not just naive Langevin or Hamiltonian Langevin dynamics (as is done in Sale and Zhang), but natural gradient Hamiltonian Langevin dynamics, which can enable faster convergence to the target distribution.

**Strengths:**

This paper is timely, and its headline premise - that an efficient sampling algorithm naturally emerges from a standard model for cortical circuits - is clearly of interest to a broad computational neuroscience audience.

**Weaknesses:**

The biggest weakness of the paper is that, contrary to what the title and the abstract would lead the reader to believe, the authors do not actually show that their model implements full Riemannian Langevin dynamics. Rather, they show that the circuit can approximate the *diagonal* of the inverse of the Fisher information matrix, rather than the full matrix inverse. The authors try to present this as a feature rather than a bug by saying that the diagonal approximation is commonly used in machine learning, but I fundamentally think that it is not appropriate to say that the network is performing emergent Riemannian MCMC in that case (I would also note that most of the cited ML applications involve non-Gaussian distributions with non-constant Fisher information matrices, for which estimating the Fisher is much more difficult than in the Gaussian case). This is a crucial conceptual feature of Riemannian MCMC that the authors miss even when they introduce the idea of using natural gradients around equation (5): not only the size, but the structure, of the sampling noise is adapted to the geometry of the problem. This discrepancy also raises the need for some more experimental tests, which I detail under **Questions**.

On the whole, I do not see this paper as being a sufficient conceptual advance relative to Sale and Zhang - which already shows evidence for Hamiltonian-like acceleration - given that the result is restricted to problem-adapted step sizes rather than adaptation of the structure of the metric, i.e., of the principal axes of variance of the sampling noise. I thus disagree with the overall tone ("remarkably", "unprecedentedly", etc).

**Questions:**

- What is the relationship of this manuscript to https://openreview.net/forum?id=BpBW4gJofo, also submitted to ICLR, with which it overlaps substantially? Even the figures look substantially similar, and the model is, so far as I can tell, precisely the same. Compare, for instance, eq.  (15) of this submission with eq. (17) of that submission. There the presentation is in terms of "adaptive step sizes" rather than approximate natural gradient, but the setting and overall scope of results are fundamentally the same. Some clarification here is required.

- Given that the circuit implements a diagonal approximation to the Fisher matrix, I think a few more experimental probes are required. Figure 3A shows convergence rates in KL over time, but I think it is critical to test how this depends on the dimensionality and structure of the target distribution. I would expect the gap between the full Riemannian MCMC and the diagonal approximation to grow as the problem dimensionality increases and also as the different dimensions become more correlated. Looking at the cited paper by Masset and colleagues (NeurIPS 2022) that is the main related work to which the authors compare their results, those authors showed that both of those factors lead to slowdown both in linear rate networks and in spiking networks that approximate the rate dynamics.

- The decomposition in eq. (15) seems non-standard relative to the usual way of decomposing the dynamics of a multivariate OU process into symmetric and skew-symmetric parts, as is done in the cited work of Ma, Fox, and colleagues (NeurIPS 2015). Could you also present the dynamics in that form?

- How does the accuracy of sampling depend on the number of neurons used? From a neuroscience perspective, this is important as it determines the minimum size of each circuit module in the setup for sampling multi-dimensional distributions that the authors present. So far as I can tell, this is not probed systematically.

**Details Of Ethics Concerns:**

Submission https://openreview.net/forum?id=BpBW4gJofo overlaps substantially with this paper. See my first Question for details.

---

> ### Author Response · Authors · 2025-11-21
>
> ### The reply to the weakness raised by the reviewer
> Our paper has significant advance and novel algorithmic insight compared with Sale 2024. Please refer to the section “The novelty of the present paper compared with Sale 2024" in Global Official Comments. Please also refer to the Generalization section there. In the revised manuscript, we add a proof-of-concept example by connecting our NG Gaussian sampling recurrent circuit with deep generating models, proving the possibility of generalizing our circuits. And we add an Appendix Sec. G in the revised manuscript to discuss the potential mechanism of generalization.
> Regarding to the reviewer’s reply about our circuit doesn’t adapt to the structure of the metric (principle axes of the sampling variance), we believe our definition of NG sampling follows the standard convention as cited in our manuscript. As long as scaling the Langevin time constant be proportional to the FI, the sampling will be in a space with metric as the FI. The details are presented in Appendix Sec. B.2. We are happy to provide more explanations in the following discussions with the reviewer to clear up the confusion.
>
>
> ### The relationship of this manuscript to the other paper
>
> We have checked the paper mentioned by the reviewer. And we believe the two papers has substantial differences and thus our paper doesn’t violate the research integrity issues (e.g., plagiarism, dual submission)  policy and doesn’t have ethical issue.
> Although both papers study the same circuit model, and the other paper also briefly mentions the adaptive sampling size, the two papers study different computational tasks and have different focuses. Our paper comprehensively studies on how the network realizes NG sampling of __static__ univariate and multivariate posteriors, and identifies the computational approximations in NG sampling in the network. By contrast, the other paper focuses on the sampling of __dynamic__ univariate posterior that changes over time, without presenting the multivariate posterior cases. And the adaptive step size in the other paper is more like a side note in the introduction of algorithmic section, without comprehensive results in the Result section (only briefly mentioned in original manuscript lines 367-372). Therefore, we think the two papers have substantial differences.
> Technically, the other paper only projected the dynamics onto the position mode, while the present study projected the circuit dynamics onto both position mode (Eq. 12a) and the __height__ mode (Eq. 12b). The height mode projection is the cornerstone for identifying the NG sampling in the circuit dynamics. These have yielded many novel results that are absent in the other paper:
> 1. The height $U_E$ (Eq. 3b) acts as the **dynamic time constant** of the position dynamics (Eq. 3a).
> 2. It dynamically **encodes the Fisher Information** of the probability distribution.
> 3. We analytically identify an embedded the __annealing strategy__ in the circuit (Eq. 3b and Sec. 4.3).
> 4. We identify the NG Hamiltonian sampling in the circuit by proving the precision of momentum decreases with FI, which is absent in the other paper.
> Combined, we believe there is no research integrity issue of the two papers, based on their significant differences of the computational tasks and conclusions.
>
> ### Diagonal FI approximation influence
>
> We provide a new simulation result about the influence of the convergence speed by diagonal FI approximation (Appendix Fig.A4). The target distribution doesn’t change, consistent with our theoretical prediction.
>
> ### The decomposition in eq. (15)
> Our decomposition serves as substantially different purposes with the one in Ma 2015. The decomposition of Ma 2015 (his Eq. 3) is to decompose the precondition matrix of a Langevin dynamics into symmetric and skew-symmetric parts, corresponding to decomposing the $\tau_L$ (becoming a matrix in multivariate case) in our Eq. 5. And the Eq. 3 in Ma 2015 is still performing Langevin sampling.
> By contrast, the decomposition in our Eq. 15 is to decompose the drift term into the part coming from Langevin sampling and the part from Hamiltonian sampling, and eventually the circuit implements a mixed sampling from Langevin and Hamiltonian. This decomposition is not canonical in ML society, while it is more like the strategy employed by neural circuit dynamics, in that similar decomposition was also used in an earlier computational neuroscience study (Aitchison, Plos Comp. Biol. 2016, Eqs. 13 – 17).

---

> ### Author Response · Authors · 2025-11-21
>
> ### The number of neurons
>
> More neurons lead to higher neuronal density $\rho = N/L$ in covering the stimulus subspace with width $L$. Based on our derived subspace dynamics (Eqs. 3a-3b), their coefficients $U_{XY} \propto \rho w_{XY} R_Y$, which implies a higher $\rho$ can be compensated with a lower connection weight $w_{XY}$ and then the sampling dynamics will remain statistically the same.
> Without scaling $w_{WY}$ with $\rho$, a larger $\rho$ will increase the likelihood precision, as suggested from $\Lambda$ in Eq. 8. And then the sampling time constant $U_E$ will also increase, as suggested from Eq. 3b.

---

> > ### Comment · Reviewer_BBUn · 2025-11-21
> >
> > Thank you for your response.
> >
> > I do not follow your argument regarding Riemannian MCMC - an approximation to the Fisher metric by definition does not realize the true natural gradient. Moreover, as your results in Appendix A.4 illustrate, there's a dimension-dependent performance gap between the diagonal approximation and using the full Fisher. Can you elaborate on what you mean here?
> >
> > I also do not find your discussion of decompositions of the dynamics convincing - the point of Ma et al. (2015) is precisely to show how Hamiltonian and Riemannian dynamics can be unified as special cases of a more general framework. Indeed, Section 3.2 of their paper shows how to introduce Hamiltonian dynamics while using the Fisher metric.

---

> > > ### Author Response · Authors · 2025-11-24
> > >
> > > > I do not follow your argument regarding Riemannian MCMC - an approximation to the Fisher metric by definition does not realize the true natural gradient. Moreover, as your results in Appendix A.4 illustrate, there's a dimension-dependent performance gap between the diagonal approximation and using the full Fisher. Can you elaborate on what you mean here?
> > >
> > > If we understand correctly, the reviewer refers to an exact natural gradient (NG) and then any approximation of the Fisher information (FI) will deviate the natural gradient.
> > > By this standard, our theoretical result suggests the canoical neural circuit can only realize the exact NG when it samples a 1D likelihood without recurrent E weight, i.e., the red line with $w_{EE}=0$ in Fig. 1E.
> > > Then, the NG will be deviated or approximated in the cases of 1) a non-zero $w_{EE}$ acting as a regularization in FI inversion (Sec. 4.2);  and 2) the diagonal FI matrix approximation in coupled circuits for 2D posterior sampling (Sec. 5). This is supported by our math analysis and the numerical results, including the new Appendix Fig. A4.
> > >
> > > Nevertheless, our criteria of claiming the NG in canonical circuits is inclusive to the __approximated__ cases in estimating FI, which is the convention in many previous papers, e.g., Liu et.al. NeurIPS 2024; Wu et.al. NeurIPS 2024; Masset, NeurIPS 2022.
> > >
> > > In practice, the diagonal FI matrix approximation will be a trade-off between computational efficiency and cost. That is, estimating the full FI matrix is computationally expensive and in most of cases the off-diagonal terms are probably small. Although the sampling with known full FI matrix is faster than the one with diagonal FI approximation(the gap in Fig. A4), when combining the high computational cost associated with full FI matrix estimation, the total cost of the full FI method may exceed the one of the diagonal FI.
> > >
> > > As a theoretical neuroscience study, our main goal is to understand the algorithm embedded in neural circuit dynamics. And our study provides exact, analytical mapping from nonlinear circuit dynamics to sampling algorithms, which is essential to develop  biologically plausible and interpretable deep network models. Even if we identified the circuit employs approximated method in estimating FI, the result itself is novel in the field and gains our understanding of neural algorithm. In this sense, we don't treat the approximation strategy in circuit is a failure of our result, but instead it is critical for us to 1) know the computational limit of neural circuits, 2) understand the bias of cognitive computations, and 3) motivate the field to update biologically plausible neural circuit models to improve their computational performance.
> > >
> > > Since the reviewer has this concern, we are thinking of the possibility of updating our title into something like "Approximate Natural Gradient Bayesian Sampling Emerges in Canonical Cortical Circuits", if it is encouraged by the reviewer. We are also open to other suggested titles.

---

> > > ### Author Response · Authors · 2025-11-24
> > >
> > > > I also do not find your discussion of decompositions of the dynamics convincing - the point of Ma et al. (2015) is precisely to show how Hamiltonian and Riemannian dynamics can be unified as special cases of a more general framework. Indeed, Section 3.2 of their paper shows how to introduce Hamiltonian dynamics while using the Fisher metric.
> > >
> > > We realize there may be a misunderstanding regarding the purpose of Eq. 15 versus Eq. 16, and how they relate to the Ma et al. (2015) framework. We are sorry that our reply above had an error in describing the Eq. 3 in Ma 2015.
> > > After careful checking the equations in Ma 2015, we are confident that the NG Hamiltonian sampling in our circuit is mathematically standard by having the same form as the Sec. 3.2 in Ma 2015.
> > >
> > > In our paper, Eq. 15 describes the governing equation in the subspaces of neural circuit dynamics, which is spanned by the excitatory bump position ($z_E$) and the SOM bump position ($z_S$). The decomposition in Eq. 15 (splitting the drift into "Langevin" and "Hamiltonian" ) is a functional decomposition to bring in momentum $p$.
> > >
> > > Once we transform Eq. 15 into the stardard form in Eq. 16, with the key mechanism of defining the momentum $p$ as a mixture of $z_E$ and $z_S$ (Eq. 15a, orange terms, check Appendix E for details of transformation), we can decompose the Eq. 16 into the Ma et al. (2015) framework. Specifically, the drift matrix in our Eq. 16 can be decomposed into a diagonal matrix $\mathbf{D}$ and a skew-symmetric matrix $\mathbf{Q}$ as in Sec. 3.2 in Ma 2015:
> > >
> > > $$ \mathbf{M} = \begin{bmatrix} \alpha_L \lambda_z (\tau U_E)^{-1} & -\beta_E \Lambda^{-1} \\\\ \beta_E \Lambda^{-1} & \tau_E \beta_p \beta_E \Lambda^{-1} \end{bmatrix} = \underbrace{ \begin{bmatrix} \alpha_L \lambda_z (\tau U_E)^{-1} & 0 \\\\ 0 & \tau_E \beta_p \beta_E \Lambda^{-1} \end{bmatrix} }\_{\mathbf{D}} + \underbrace{\begin{bmatrix} 0 & -\beta_E \Lambda^{-1} \\\\ \beta_E \Lambda^{-1} & 0 \end{bmatrix}}\_{\mathbf{Q}} $$
> > >
> > > In this way, the sampling dynamics embedded in our circuit has the conventinoal form as Sec. 3.2 in Ma 2015.
> > >
> > > When comparing the coefficients the matrix $\mathbf{M}$'s coefficients in our circuit with Ma 2015, we found there are two differences.
> > > 1. $\mathbf{D}_{11}$ in our circuit is non-zero while the one in Ma 2015 is zero, because we found the circuit employes a mixture of Langevin and Hamiltonian sampling.
> > >
> > > 2. When comparing the $\mathbf{M}$'s coefficients or the whole drift term with the order of Fisher information $G(z)$, we realize our defined NG Hamiltonian had different orders with Ma 2015, while ours (Eq. 7) follow the convention defined in Girolami 2011 (their Eqs. 13-15).
> > > In detail, the drift term in the Sec. 3.2 in Ma 2015 is (we change his $\theta$ into our notation $z$, and momentum $r$ into our notation $p$)
> > > $$
> > > \mathbf{M} \cdot \nabla H(z) =
> > > \begin{bmatrix} 0 & - G^{-1/2} \\\\ G^{-1/2} & G^{-1} \end{bmatrix}
> > > \begin{bmatrix} \nabla_z U(z) \\\\ p \end{bmatrix}
> > > $$
> > > where $U(z)$ is the negative of the log-likelihood.
> > > In contrast, the Hamlitonian component (setting $\mathbf{M}_{11}=0$ to remove the Langevin compoennet) embedded in our circuit model is
> > > $$
> > > \mathbf{M} \cdot \nabla H(z) =
> > > \begin{bmatrix} 0  & - \mathcal{O}(1) \\\\
> > > \mathcal{O}(1) & \mathcal{O}(G) \end{bmatrix}
> > > \begin{bmatrix} \nabla_z U(z) \\\\ G^{-1} p \end{bmatrix}
> > > $$

---

> > > > ### Comment · Reviewer_BBUn · 2025-11-24
> > > >
> > > > Thanks for your reply! This helps clarify for me how the structure of the dynamics here relate to those in Ma et al.
> > > >
> > > > With regards to the semantics of calling something a natural gradient, my chief concern here is that your network is restricted to diagonal approximations to the Fisher information matrix. As your results show, this results in a substantial performance gap between this approximation and true natural gradient for Gaussians with non-diagonal covariance. I view this as a major limitation. Also, as you're considering a Gaussian setting where the Fisher is constant, I don't follow appeals to computational cost of computing it. Can you elaborate on what you mean here?

---

> > > > > ### Author Response · Authors · 2025-11-25
> > > > >
> > > > > As a theoretical neuroscience study, our primary objective is to identify the computational algorithms of biological neural networks. Whether and how Natural Gradient sampling is realized in the brain has remained largely unknown.
> > > > > To provide algorithmic insight, we start from a widely used biologically plausible recurrent circuit model with types of interneurons in computational neuroscience research, and perform rigorously theoretical analysis to identify the circuit algorithm. And the diagonal FI approximation is one of our discovered results emerging from circuit dynamics (rather than we purposely design), which represents a novel and significant conceptual advance in the field.
> > > > >
> > > > > We admit the pursuit of more powerful computation is endless in both artificial and biological neural networks. Although in principle it is possible to design recurrent networks to realize the full FI, whether such networks are biologically plausible remains a big issue.
> > > > > It also remains unknown whether the brain can utilize the full FI for natural gradient, which is still an unexplored topic in computational neuroscience research. Anyhow, studying whether and  how biological neural circuits realize full FI for natural gradient is an important extension and forms our future study. But designing a new class of recurrent networks that 1) remains biological plausibility, 2) capable of full FI, and 3) perform rigorously theoretial analysis is a substantial undertaking that lies beyond the scope of the present study and needs a series of studies.
> > > > >
> > > > > At last, even if we also hope the neural circuits in the brain realize full FI for natural gradient, it is also possible that neural circuits in the brain cannot realize that as everything has limitations, which is still an open question in the field.
> > > > >
> > > > > > Also, as you're considering a Gaussian setting where the Fisher is constant, I don't follow appeals to computational cost of computing it. Can you elaborate on what you mean here?
> > > > >
> > > > > First, even when the FI is constant given a Gaussian posterior, we find the canonical recurrent neural circuits with fixed synaptic weights can automatically adjust their sampling step size with various posterior uncertainties, a remarkable property that has not been emphasized in previous studies especially artificial network studies.
> > > > >
> > > > > For Gaussian cases, we agree it is relatively easier to estimate the non-diagonal elements of a FI matrix. In addition, the FI matrix in Gaussian cases remains constant over the state space, and then the NG sampling only needs to invert the FI matrix once, avoiding the need of FI matrix inversion in every time step.
> > > > >
> > > > > Nevertheless, the diagonal FI approximation probably scales better with more complex distributions especially the ones whose FI depends on local states. That's why many practical algorithms, e.g., Adam or RMSProp, utilize diagonal approximation rather than full FI matrix.
> > > > > This simplifies the architecture and dynamics of canonical circuit (the circuit with full FI inversion may require complex wiring that is not biologically plausible), and enhance the generalization of our canonical circuit with inherent diagonal FI approximation to more complex distributions in the future, as many previous studies suggest the neural circuits can sample non-Gaussian and even long-tail distributions whose FI depends on the location in the state space. Details regarding the generalization of our circuit can be found in the global reply.

---

### Official Review · Reviewer_g6vX · 2025-11-08

**Soundness:** 3
**Presentation:** 3
**Contribution:** 2
**Rating:** 4
**Confidence:** 3

**Summary:**

This paper extends the work of Sale & Zhang (2024) on Bayesian sampling in canonical cortical circuits to show that these circuits naturally implement natural gradient (NG) sampling algorithms. The authors propose that the canonical circuit, consisting of excitatory neurons and parvalbumin/somatostatin interneurons, can implement both natural gradient Langevin and Hamiltonian sampling. The key mechanism is that the total activity of E neurons (bump height $U_E$​) monotonically increases with the posterior's Fisher information, which automatically adjusts the sampling step size based on the local geometry of the posterior distribution. The authors demonstrate through theoretical analysis that the circuit implements NG Langevin sampling in the reduced E+PV circuit, adding SOM neurons enables NG Hamiltonian sampling, non-equilibrium dynamics during the transition from resting to evoked states further accelerates sampling through an intrinsic annealing strategy, and coupled circuits can sample multivariate posteriors using diagonal Fisher information matrix approximations analogous to techniques in machine learning.

**Strengths:**

1. Originality: the paper makes a theoretical contribution by identifying that canonical cortical circuits naturally implement natural gradient sampling, extending beyond the naive Langevin and Hamiltonian sampling identified in Sale & Zhang (2024). The connection between E neuron bump height and Fisher information provides a novel functional interpretation of circuit dynamics. The identification of computational approximations in the circuit (regularization via recurrent connections, diagonal FIM approximation) that parallel ML techniques is conceptually interesting.
2. Quality: the mathematical analysis is rigorous and follows established methods for analyzing continuous attractor networks. The authors provide detailed perturbative analysis, eigenmode decomposition, and explicit mappings between circuit dynamics and NG sampling algorithms. The derivations connect circuit parameters (bump height $U_E$​) to Fisher information and sampling step sizes in a principled way.
3. Clarity: the paper is generally well-structured. The progression from naive sampling to NG sampling is clearly explained, and the figures are effective in illustrating the main concepts, in particular showing how bump height scales with Fisher information and how this determines sampling time constants. The supplementary materials provide thorough derivations.
4. Significance: the potential unification of natural gradient sampling within the canonical circuit architecture is conceptually appealing. The model makes concrete predictions about the relationship between E neuron population activity, Fisher information, and sampling efficiency. The connection to ML approximation strategies (regularization, diagonal FIM) bridges neuroscience to machine learning.

**Weaknesses:**

1. I am not sure about the computational necessity of sampling in the circuit model if not to encode posterior uncertainty. The authors state that "a single snapshot of $r_F$ parametrically conveys the whole stimulus likelihood" (Eq. 8), meaning a population vector readout is sufficient. Given this, it's unclear what computational advantage NG sampling provides over simpler population coding schemes like probabilistic population codes (PPC), which can also perform Bayesian inference with linear readouts but without the complexity of maintaining sampling dynamics. Since the posterior uncertainty is entirely determined by the feedforward input rate $r_F$ (which controls likelihood precision $\Lambda$), the neural variability does not represent posterior uncertainty in the way that sampling-based models typically propose. For the Gaussian likelihoods and uniform priors assumed in this framework, deterministic inference methods (like direct computation of the posterior mean and variance) would be exact and more efficient. The authors do not provide quantitative comparisons of computational costs, convergence speed, or accuracy against such alternatives.
2. Moreover, restrictive assumptions limit the generality of this approach. The framework relies heavily on several assumptions. First, the model assumes Gaussian feedforward tuning curves (Eq. 1e) leading to Gaussian likelihoods (Eq. 8). However, real sensory likelihoods are often non-Gaussian and multimodal, and one of the purported strengths of sampling-based approaches is that they can represent arbitrary distributions. The authors do not address how the circuit would handle non-Gaussian inference problems. Second, the model assumes a 1D ring attractor, and its specific eigenmode structure is essential for the perturbation analysis. While the authors show a 2D extension (Fig. A4) that couples two ring attractors, this is still a rather restrictive latent structure which presumably not all canonical circuits possess, and how to scale to higher-dimensional feature spaces without this specific structure remains unclear. Finally, the analysis uses uniform priors throughout. This eliminates one of the key computational challenges of Bayesian inference, which is to show that the prior can reflect the statistics of its inputs. The authors do not demonstrate that the circuit can implement informative non-uniform priors or flexibly switch between different prior distributions.
3. The circuit already receives critical information that undermines claims of adaptive inference. Specifically: the feedforward input $r_F$ directly encodes the likelihood mean $\mu_z$ and precision $\Lambda$ (Eq. 8), meaning the posterior parameters are essentially pre-computed and fed into the circuit rather than inferred. Moreover, while the authors claim the circuit "adaptively adjusts the sampling step size," this adaptation is entirely driven by the feedforward input intensity $R_F$, which is externally provided rather than computed by the circuit itself. The recurrent weights ($w_{EE}$, $w_{SE}$, etc.) are also precisely tuned to match the required sampling parameters (Eqs. 11, Fig. 4E), but the authors do not mention how these precise weight configurations would be acquired or whether synaptic plasticity could maintain them.
4. The authors briefly mention that deterministic inference circuits require "complicated nonlinear functions," but this claim is specific to their problem structure (Gaussian distributions, linear-Gaussian dynamics). However, they do not provide quantitative comparisons of computational costs relative to deterministic approaches, convergence speed relative to standard (non-NG) sampling, accuracy trade-offs between their approach and alternatives, and performance relative to the single previous natural gradient sampling circuit study (Masset et al., 2022). The claim that NG sampling provides advantages over naive sampling is not substantiated with systematic quantitative comparisons.
5. There is limited biological justification for key mechanisms in the model. While the circuit architecture is based on known connectivity patterns, the assumption that recurrent E weights ($w_{EE}$) act as a regularization parameter (analogous to $\alpha$ in Eq. 5) is mathematically convenient but lacks biological justification. How would the circuit "know" to set this weight to prevent numerical instabilities in Fisher information inversion? Moreover, the non-equilibrium annealing strategy is presented as an "emergent property," but the functional advantage of this particular annealing schedule over other possible dynamics is not demonstrated.

**Questions:**

1. How would the circuit handle non-Gaussian likelihoods, which are common in real sensory processing? Can you provide numerical experiments or extensions showing the framework handles cases where the Gaussian assumption breaks down?
2. Given that the likelihood can be read out with a population vector (linear decoder) from $r_F$, what specific computational advantages does NG sampling provide over probabilistic population codes (PPC) or direct computation of posterior parameters? Can you provide quantitative comparisons (e.g. in terms of inference accuracy or speed, or computational cost)?
3. For Gaussian likelihoods and uniform priors, the posterior is also Gaussian with analytically computable parameters. What is the computational advantage of approximating this via sampling when exact solutions are available? Under what conditions would sampling-based inference be preferred?
4. How would the circuit acquire the precise weight configurations required for NG sampling (e.g., relationships in Eqs. 11, 14, Fig. 4E)? Do you propose these synaptic weights are learned, and if so, through what learning rule?
5. The claim that recurrent E input acts as regularization (like $\alpha$ in Eq. 5) is interesting, but how does the biological circuit "know" to set $w_{EE}$ to prevent numerical instabilities in Fisher information inversion? What mechanisms would maintain this relationship as environmental statistics change?
6. In multivariate cases, the circuit uses diagonal FIM approximation rather than full FIM. Can you quantify the loss in performance relative to full FIM?
7. How does the NG sampling in your circuit compare to naive Langevin/Hamiltonian sampling, deterministic inference, and the NG sampling circuit in Masset et al. (2022)? I would appreciate specific metrics like convergence time, sample efficiency, and accuracy.
8. The non-equilibrium annealing is claimed to accelerate sampling (Fig. 2K), but by how much compared to equilibrium sampling? How does this depend on circuit parameters?
9. How sensitive is the NG sampling to mismatches between the assumed circuit parameters and the true parameters? For example, what happens when $w_{EE}$ deviates from the value required for proper regularization, or when Fisher information is estimated incorrectly?
10. Can the circuit handle time-varying Fisher information (e.g., if the feedforward input statistics change over time)? How quickly can the circuit adapt its sampling strategy to new posterior geometries?

---

> ### Author Response · Authors · 2025-11-21
>
> Thanks for the positive comments from the reviewers and the recognition of our theoretical insight in understanding how canonical circuits implement NG sampling. Below we reply to your weaknesses and questions in the same order shown in your review.
>
>
> ### The necessity of sampling in the circuit model if not to encode posterior uncertainty
> We thank the reviewer for raising this important point. It appears the sampling is computationally unnecessary for the Gaussian cases with exact solutions, while the sampling can simplify the complexity of circuit implementation compared with deterministic algorithms and it fits the neural internal variability.
> 1. Circuit implementation complexity. Simpler or exact computational algorithms don’t necessarily mean an easier neural circuit implementation. We agree reviewer’s comment that it appears computationally unnecessary to employ sampling to compute the 1D Gaussian likelihoods when they are parametrically represented by feedforward inputs. Nevertheless, when considering the circuit implementation of bivariate posterior with a non-uniform prior (Fig. 3A and Sec. 5), the deterministic network requires complex, nonlinear interactions between networks, as shown in Fig 3A in Raj & Pitkow NeurIPS 2016. This increases the complexity of neural circuit implementation. In contrast, our sampling circuit only requires linear interactions between networks.
> Regarding the neural variability in our 1D Gaussian likelihood case (Fig. 2), the variability of stimulus samples, linearly read out from neural responses, represents the likelihood variance (Fig. 1F-H), even if the feedforward input is modeled as PPC.
> 2. Neural variability. Biological neural networks are inherently noisy, and accumulating evidence suggests the brain utilizes the sampling. By contrast, the deterministic algorithm/circuit cannot explain the internal neural variability.
>
>
> ### The weakness of restrictive assumptions
> We are also interested in generalizing our circuit model to non-Gaussian distributions and non-uniform priors. Please refer to the Global Official Comments for our generalization. In addition, we added a proof-of-concept example in a new section and Fig. 5 in the revised manuscript. We consider connect our recurrent circuit with non-trivial encoder and decoder, and then the circuit can deal with non-trivial distributions. We also augmented the Discussion and Appendix (Sec. G) by discussing the theoretical mechanism of potential generalization.
>
> ### Adaptive inference and parameter sensitivity
> In the 1D likelihood case (with uniform prior) in Fig. 2, the circuit directly accesses the likelihood precision from the feedforward input due to its parametric nature. However, in the 2D posterior case where the non-uniform associative prior is stored in the coupled circuits (Fig. 3A), the circuits need to read out its internally stored prior to compute the posterior precision that in turn determine the sampling step size. The latter case makes the precision computation non-trivial in our circuits.
> Regarding to the parameter sensitivity, our circuit doesn’t need fine-tuned. Appendix Fig. A3 shows there is a line manifold in the parameter space spanned by the feedforward weight $w_{EF}$ and SOM inhibitory weight $w_{ES}$ to enable the circuit to sample correct posteriors. The recurrent E weight $w_{EE}$ doesn’t affect the equilibrium sampling distribution, while it only affects the regularization strength in FI inversion.

---

> ### Author Response · Authors · 2025-11-21
>
> ### Quantitative comparison with deterministic inference circuits
> Please refer to our reply to your 1st weakness about the nonlinear operations in deterministic circuits to infer 2D posteriors. We believe the substantial difference in required circuit architecture for deterministic vs sampling algorithms is significant enough to distinguish them especially the biological plausibility. In terms of computation, the specific form of required nonlinear operations in deterministic circuits depends on the problem structure, e.g., Gaussian likelihoods. Even when dealing with relatively simple Gaussian problems, deterministic circuits require complex nonlinear interactions. This makes us concerned about the capability of deterministic circuits to generalize to more complex distributions, especially the form of nonlinearity changes with different distributions.
>
> We are also interested in direct quantitative comparison of the computational cost and convergence speed across different circuit models. However, it is extremely challenging and even implausible sometimes, which is reflected by the fact that such a direct comparison is largely missing in previous computational studies. First, different circuit models consider different generative models and posteriors as mentioned in our Discussion, which makes a fair comparison challenging. Second, although comparing algorithms are conventional (e.g., comparing Eq. 10 with alternatives), comparing the performance of circuit implementation (comparison Eqs1a-1e with alternatives) is __infeasible__ sometimes. For example, if we want to compare the NG sampling vs non-NG at the algorithmic level, we can simply clamp the $U_E$ in Eq. 10 and run the simulation. However, in the circuit level, the NG sampling (Eqs. 3a-3b) is embedded in the automatic, high-dimensional circuit dynamics, and we don’t know how to manually clamp the $U_E$ in the circuit dynamics. Then, it requires us to develop a new neural circuit model that doesn’t implement NG, which remains largely unknown. And it is probably not worth developing a new circuit model to implement less efficient non-NG sampling.
> Certainly, we are open to hearing more detailed suggestions from the reviewer to make a fair comparison to improve our manuscript.
>
> > How would the circuit "know" to set this weight to prevent numerical instabilities in Fisher information inversion?
>
> It is beyond the scope of the current study since it is already dense, while we provide our thought here.
> The regularization is a hyperparameter, and how to adjust it remains open in neuroscience. One possibility is adjusted via the top-down modulation based on task needs. Our contribution lies in providing direct link between $w_{EE}$ and regularization.
>
> ###  The functional advantage of non-equilibrium annealing strategy
>
> Due to the space limit, we don’t present this result. One possibility is the strong PV inhibition induces bump height $U_E$ oscillation, enabling the circuit to implement cosine-profile annealing. Apart from monotonically increasing $U_E$, the oscillations of $U_E$ can implement cosine-profile annealing (briefly mentioned in line 481 in original manuscript), which can be realized by adding an explicit PV dynamics with stronger PV inhibition. The $U_E$ oscillation will benefit multi-modal distribution sampling where the increase of step size during oscillation cycle can help samples go to another mode in distributions.
>
> > How would the circuit handle non-Gaussian likelihoods, which are common in real sensory processing? Can you provide numerical experiments or extensions showing the framework handles cases where the Gaussian assumption breaks down?
>
> Please refer to the Global Official Comments for our detailed plan of generalization. We have augmented the Discussion and Appendix (Sec. G) by inserting the plan of generalization.
>
> > Given that the likelihood can be read out with a population vector, what specific computational advantages does NG sampling provide over PPC or direct computation of posterior parameters? Can you provide quantitative comparisons ...
>
> See the reply to the necessity of sampling in the Global Official Comments.
>
> > What is the computational advantage of approximating this via sampling when exact solutions are available? Under what conditions would sampling-based inference be preferred?
>
> See the reply to the necessity of sampling, and non-uniform prior in the Appendix (Sec. G2).
>
> > How would the circuit acquire the precise weight
>
> We think the synaptic weights are learned,d but investigating a specific learning rule is beyond the scope of this paper.
>
> >The claim that recurrent E input acts as regularization (like in Eq. 5) is interesting, but how does the biological circuit "know" to set to prevent numerical instabilities in Fisher information inversion? What mechanisms would maintain this relationship as environmental statistics change?
>
> See our reply above.

---

> ### Author Response · Authors · 2025-11-21
>
> ### Diagonal FIM approximation’s performance
>
> The equilibrium sampling distribution remains the same when changing time constant. In Fig3, it shows the sampling is slower for diagonal FI approximation. Also, we provide a new figure of the convergence time with the increase of dimension in Appendix Fig. A4. The converges time of full FI basically doesn’t change with the dimension, while the one of diagonal FI increases.
>
> ### Comparison with other ways
>
> 1. NG Sampling vs. Naive Sampling: The extensive empirical benchmarking of NG sampling against naive Langevin/Hamiltonian sampling is a well-established theoretical result in Amari 1998. Our core contribution is not re-proving that NG is faster, but analytically showing that the nonlinear circuit dynamics automatically implements the NG sampling. The efficiency shown in Fig. 2 are a direct consequence of this mathematical equivalence.
> 2. NG Sampling vs. Deterministic Inference: Please refer to our reply earlier. Although comparing algorithms are conventional, comparing the circuit dynamics implementing different algorithms are challenging and even infeasible sometimes.
> 3. Comparaison to Masset et al. (2022): A fair quantitative comparison is challenging because these two network different generative models.
>
> ### The non-equilibrium annealing performance
> The result is shown as the black dashed line in Fig. 2K, which is obtained by using samples generated from neural circuits when its bump height reaches equilibrium. The non-equilibrium annealing also depends on the recurrent E weight $w_{EE}$ as shown in Fig. 2K: a smaller $w_{EE}$ leads to faster sampling as long as the input intensity if not too low.
>
> ### Mismatches between the assumed circuit parameters and the true parameters
> The FI represented by the bump height $U_E$ doesn’t change the equilibrium sampling distribution (Eq. 10), but only changes the sampling speed. The feedforward weight $w_{EF}$ and SOM inhibitory weight $w_{ES}$ will change the equilibrium distribution, and Appendix Fig. A3 plots how their value determine the divergence between posterior and the sampling distribution.
>
> ### Time-varying Fisher information
> Yes. Fig. 2I-K can be viewed as an example of time-varying FI where the FI increases from zero (input is off) to a non-zero value (input is ON). The speed of the transition is controlled by the time scale of the bump height $U_E$ dynamics. Specifically, it is determined by the $\tau$ and $w_{EE}$.

---

### Official Review · Reviewer_fZPj · 2025-11-08

**Soundness:** 3
**Presentation:** 3
**Contribution:** 3
**Rating:** 6
**Confidence:** 5

**Summary:**

This paper analyses the dynamics of E-I circuits (including two classes of I neurons, PV and SOM) with ring architectures, and through extensive mathematical derivations and also numerical simulations shows that they perform particular forms of sampling (natural gradient Langevin and Hamiltonian) from simple (mostly 1D Gaussian) distributions.

**Strengths:**

The fundamental goal of relating neural circuit dynamics to (sampling-based) inference is interesting, and there is a lot of nice ideas for working out such a relationship (e.g. the overall magnitude of neural activities acting as regulator of step size, or the putative role of SOM neurons in sampling). The mathematical analyses are extensive, and the paper is generally well written.

**Weaknesses:**

1. There are a number of seemingly arbitrary choices in model construction:

- Why is PV vs. SOM cell-mediated inhibition taken to be global, divisive (in firing rates), and instantaneous vs. local, subtractive (in synaptic inputs), and finite time-scale (referring to both the time constants of the I cells, and their effects on E cells), respectively? (E.g. if anything, I would have thought PV inhibition is more local than SOM is.)

- Why are only E but not I cell synaptic inputs noisy?

2. Overall, the circuit is shown to be able to sample from a 1D Gaussian (or the 1D marginals of a 2D Gaussian, see below). Sampling becomes particularly useful for high dimensional joint posteriors, and with more complicated distributions. It's unclear if such more challenging forms of sampling can be solved by this approach (although the Discussion does mention some future directions in this regard). Specifically, the Gaussian represents a special case for NG Langevin sampling, in which the FI is constant, and as such, equivalent to just changing the (effective) time constant. So it's unclear if the circuit actually continues to accurately approximate NG Langevin in the more general case, when the FI changes as a function of the sample (again, I acknowledge the mentioning of this in the Discussion). The setup also seems to require that the latent variable is directly encoded in a Gaussian-Poisson population code, so that the likelihood is a Gaussian, whose precision scales linearly with the input firing rates. Again, it is unclear how this can be generalized to cases of practical interest, in which neither of these assumption will hold generally.

3. Based on the derivations, in order for the circuit to implement NG Langevin sampling, U_E should scale linearly with FI (Eq.11). Yet, in the simulations, when w_EE is sufficiently large, there is a strong threshold nonlinearity in the relationship between the two (Fig.2C). Furthermore, the authors suggest that U_EE (controlled by w_EE) acts as a regularizer "improving the numerical stability in inverting the FI when it is small or ill-conditioned". However, the empirical relationship between U_E and FI (Fig.2C) seems to be such that specifically at the small FI values, where regularization is supposed to be useful, there is no difference between small and large w_EE values. Indeed, the results shown in Fig.2F barely show any advantage for a larger w_EE in the small FI regime (and it's somewhat conspicuous why w_EE/w_c is not shown). All this leaves it unclear what the role of w_EE is in sampling, and whether it really implements the kind of regularization the paper proclaims. This is a problem, because this seems to be the only function suggested for recurrent excitation in the circuit, and otherwise a purely feedforward circuit seems best (see e.g. Fig.2E).

4. It is unclear what's the advantage of the "non-equilibrium" "annealing" strategy of the circuit at stimulus onset, shown in Fig.2I-K. First, large differences in the steady state bump heights (at different values of w_EE; Fig.2J) translate to minimal differences in sampling speed gains (Fig.2K). In fact, if anything, it seems that the case when the bump height barely grows at stimulus onset (w_EE/w_c=0) results in slightly faster sampling. I couldn't find what U_E the "equilibrium NG" sampler used, but I suspect it used the large value corresponding to the stimulus being on. What about an "equilibrium NG" sampler that always uses the low U_E corresponding to the stimulus being off? If I am right, there is no special advantage to the "non-equilibrium" "annealing" the authors focus on — there is simply an advantage of using low U_E as long as possible.

I also found the terminology here somewhat fanciful (in that it made the effects that are described here sound more fancy than they really are). The term "annealing", in the context of sampling, typically refers to procedures that make the sampler sample progressively different target distributions (or an optimizer to optimize progressively different objective functions, as in simulated annealing). This is not the case here — the sampler samples from the same posterior distribution throughout the period of "annealing", just with different time constants. The term "non-equilibrium" dynamics usually refers to autonomous dynamical systems. This is also not the case here — the autonomous dynamics of the system here itself changes during the "non-equilibrium" phase of the experiment because the inputs to the system change.

5. The bivariate setup is a little confusing. The main text makes it sound as if the combined circuit sampled from the joint posterior, and a paragraph and Fig.A2A-B is devoted to explaining the properties of the correlated prior the circuit implements. In turn, such a correlated prior is interesting because it also makes the posterior correlated (especially because the likelihoods are independent). However, then Fig.3 (and its caption) emphasizes how each module samples the corresponding marginal posterior. Indeed, I found no demonstration that the joint posterior is correctly sampled (unless the precisions shown in Fig.A2D are somehow related to joint precisions). But then how is this more useful than two decoupled circuits, already covered by the preceding sections?

A more minor point is that Eq.13 suggest that the prior implemented by cross-module weights is a bivariate Gaussian. But then the text states that it stores "an associative (correlational) stimulus prior with each marginal uniform" — it would be useful to point out that this is consistent with a bivariate Gaussian as a degenerate case.

6. Taking together the concerns above, there is something slightly odd about what the proposed neural circuit achieves computationally in the end: it samples from a likelihood that is already parametrically represented (in a very easily readable form) in the input. If my interpretation of the bivariate case is correct (point 5), there is no combination with a nonuniform prior, or with some other likelihood. If this is the case, then there seems to be no additional benefit to the operation of this circuit compared to what its input already provides.

7. The role of attractors in the circuit dynamics is confusing. The starting point for all the mathematical analysis is based on the existence of attractor states in the network. However, if these are truly attractor states, then they persist even in the absence of a stimulus — this is a problem because if the stimulus disappears (or more generally, changes), we would not want the circuit to maintain its representation of the posterior that was based on the (previous) stimulus. Indeed, based on Table 1, it seems that recurrent weights were chosen to be smaller than w_c (0.8w_c), which is "the smallest value that allows the network to maintain persistent activity even when there is no feedforward input" — i.e. the network is *not* in the parameter regime in which it has attractors.

So, it is unclear, whether the presented networks do or do not have attractor states, and in the former case, how they can usefully perform sampling, and in the latter case, how the mathematical derivations serve their understanding.

***

Minor issues:

l.59: "linear dynamics of Langevin and Hamiltonian samplings [...] used in machine learning (ML) research" Is it really true that parctical sampling algorithms used in ML research use linear dynamics?
l.205: "We will leave the subjective prior p(s) unspecific": "p(s)" → "p(z)"
l.215: "is resulted from" → "results from"
l.258: "we investigate the how the" → "we investigate how the"
l.260: "It is because" What is because?
l.269: "the circuit with fixed weights flexibly sampling likelihoods": "sampling → "samples"
l.308: "sampling step size will gradually decreases": "decreases" → "decrease"
l.350: "denots" → "denotes"
l.357: "To ease of understanding": "To" → "For"
l.410: "satisfys" → "satisfies"
l.465: "flexibly" → "flexibility"

Fig.2: the orange-red colors are barely distinguishable
Fig.2F: what's the black dashed line?
Fig.2H: what are the solid vs dashed lines?
Fig.A2: what is Lambda_s (I couldn't find its definition)?

**Questions:**

1. Is there any biological evidence to back up the modeling choices mentioned in point 1 above?

2. Is there any evidence that the network is able to sample from non-Gaussian posteriors, and in cases in which the likelihood is not given by a Gaussian-Poisson population code?

3. Is there any robust evidence (beyond what is currently shown in Fig.2C) that non-zero w_EE has a useful functional role in the circuit?

4. Is there any evidence that "annealing", rather than just a generally low value for U_E, is specifically useful for sampling after stimulus onset?

5. Is there any evidence that the samples produced by the network faithfully represent the correlations under a joint posterior distributions? If so, how can they usefully perform sampling, and if not, how do the mathematical derivations serve their understanding?


6. Is there a way to demonstrate that the network represents a posterior which cannot be trivially decoded already from its input?

7. Are there attractors in the intrinsic dynamics of the network?

---

> ### Author Response · Authors · 2025-11-21
>
> Thanks for the reviewer’s positive comments about the math rigor, writing, and algorithmic insight of our paper. We will proofread our manuscript carefully to solve the grammatical errors. Below, we will reply to your weaknesses and questions one by one.
>
> ## Biological evidence of PV and SOM
> The page limit doesn’t allow us to elaborate on the model choice in the manuscript. Our circuit model is inherited from Sale, NeurIPS 2024. Below, we summarize the rationale for constructing our model.
> 1. Divisive PV vs subtractive SOM
> As a post hoc test, our model reproduces the divisive and subtractive modulations of E neuron tunings from PV and SOM neurons, respectively (Fig. A1 A-B) that are qualitatively similar to the results in Wilson, Nature 2012, supporting the biological plausibility of our model. The inclusion of PV in the divisive normalization (DN) is directly inspired by previous studies claiming the DN is formulated by PV neurons, which probably through shunting inhibition.
> 2. Global PV vs local SOM
> In the anatomical locations on the cortical surface, SOM reaches a larger region than PV, as suggested by the reviewer. Nevertheless, our circuit model is defined on the feature space like orientation. And PV’s orientation tuning is much weaker than SOM (Adesnik, Nature 2012), which can be reproduced by our model setting that PV is globally driven by all E neurons with homogeneous weights. By contrast, SOM neurons have significant orientation tuning, suggesting they only monitor a portion in the orientation space and is more local than PV in terms of orientation.
> 3. Instantaneous PV vs finite time-scale SOM
> The PV neurons are fast-spiking and may have a faster time scale than E neurons. To simplify our math analysis, we consider the PV dynamics is instantaneous without temporal dynamics, although having an explicit dynamics will enrich the annealing strategy in the circuit (see below reply to annealing). Also, for simplicity, we consider the time constant of SOM is the same as E, although actual SOM may be slower than E. This simplicity doesn’t affect the sampling in the circuit. The change of SOM time constant can be compensated by the inhibitory weight from SOM to E neurons, without changing the equilibrium distribution.
>
>
> ## Why are only E but not I cell synaptic inputs noisy?
>
> This is a simplification of our circuit model to facilitate math analysis. Including noises in I neurons won’t change the circuit sampling substantially and can be compensated by re-setting the weights. For example, adding noises to SOM neurons will bring an extra noise term into Eq. 15b, which increases the $\sigma_p^2$ in Eq. 16. Then we only need to readjust the purple coefficient in the 1st matrix on the RHS in Eq. 16 to enable the circuit sample correct posteriors.
>
> ## 1D marginals of a 2D Gaussian and non–Gaussian case
>
> Firstly, 2D posterior sampling in Fig. 3 not simply sample two 1D marginal distributions. When combining/concatenating the samples individually read out from each network (Fig. 3C), the 2D samples (Fig. 3D, the colored trajectory) approximate posteriors. Our analysis shows the coupled circuits store a non-uniform associative prior (Eq. 13). In addition, even if the FI of a gaussian is constant, our proposed circuit with the same weight can automatically scale the sampling time constant with posteriors precision, as shown in 1D likelihood sampling in a single recurrent circuit (Fig. 1D). We further add an Appendix Fig. A2E in the revised manuscript showing the posterior (with non-uniform prior) sampling in coupled circuits also automatically change its sampling step size with posterior FI. This is an elegant property of our circuit model, which has not been reported in previous theoretical results.
> Secondly, we acknowledge the reviewer's interest regarding the NG sampling of non-Gaussian and more complex distributions. We are also interested in this generalization. Please refer to the Generalization section in the Global Official Comments and we add a new section of a proof-of-concept example in the revised manuscript.
>
> ## legend about recurrent weight and the role of recurrent weight
>
> Figure 2F shows the $w_{EE}/w_c$ ratio. The colormap of Fig.2E is the same as Fig.2C. Fig.2J&K has the same color map with Fig.2F. So, the solid line in Fig 2 K is the network results with or without recurrent weight. And we apologize for the confusing legend, and we will update it in the revised manuscript soon.
>
> Notably, the convergence time for the model with recurrent weights is roughly half of that for the model without them.
> In this paper, we primarily discuss the role of the recurrent weight within the context of the natural gradient, while it is no doubt that recurrent weights are fundamental to other circuit functions, e.g., evidence accumulation in decision-making, support persistent activity in working memory, remove input noises, etc.

---

> ### Author Response · Authors · 2025-11-21
>
> > Is there any evidence that "annealing", rather than just a generally low value for U_E, is specifically useful for sampling after stimulus onset?
>
> The equilibrium sampling in Fig. 2K use the samples generated from the network when its bump height $U_E$ reaches the equilibrium state (details in Appendix F2), and therefore the network uses smaller step size. If we just simulate the sampling $z_E$ dynamics in circuit’s subspace (Eq. 10) and __manually__ set the $U_E$ to be lower than its equilibrium and fix it over time, the equilibrium sampling in this case can be faster than the non-equilibrium curves (the blue line in a new Appendix Fig. A5). Hence, the reviewer’s conjecture is correct. Note that the $U_E$ value cannot be lowered to the value corresponding to stimulus being off. In that case, the step size will be too large and then slow down sampling (the case shown at the left-most part in Fig. 2F), or even leads to instability.
> Although we can simulate $z_E$ dynamics with manually lowering $U_E$, we’d emphasize this manual approach serves as to help us understand the circuit computation but is probably __infeasible__ in simulating the neural circuit dynamics (Eqs. 1a-1e). The __automatic__ circuit dynamics wrap up the $z_E$ and $U_E$ dynamics (Eq. 3a-3b), and evolves by itself using its intrinsic dynamics of adjusting $U_E$ (Eq. 3b). That means we cannot manually clamp $U_E$ in the full circuit dynamics as a low value as we did in identified subspace dynamics (Eq. 3a-3b). Certainly, we welcome the reviewer’s suggestion if there is an approach of doing that. In this sense, the non-equilibrium circuit sampling still has its value in speed up circuit sampling, especially in contrast to most previous studies focusing on circuit samples from equilibrium.
> Apart from monotonically increasing $U_E$, the oscillations of $U_E$ can implement cosine-profile annealing (briefly mentioned in line 481 in the original manuscript), which can be realized by adding an explicit PV dynamics with stronger PV inhibition. The $U_E$ oscillation will benefit multi-modal distribution sampling where the increase of step size during oscillation cycle can help samples go to another mode in distributions.
>
>
> ### Reply to terminology
>
> We thank the reviewer for the comment on the terminology. We respectfully clarify that our use of 'annealing' is intended in the broader, physical sense, rather than as a strict implementation of 'simulated annealing' for optimization. In physical annealing, cooling a system alters its internal dynamics like reducing kinetic energy to allow it to find a stable, low-energy state. Our usage is directly analogous and follows a precedent in the literature (Ahn et al., ICML 2012), where 'annealing' refers to the scheduled reduction of a sampler's dynamic parameters including its step size or noise level. This guides the sampler from a fast, exploratory ("hot") phase to a stable, convergent ("cold") phase. This 'cools' the dynamics of the sampler itself, not the (fixed) target distribution.
> Additionally, our circuit model can sample progressively different target distributions during annealing as you mentioned. For 2D posterior sampling in coupled circuits (Fig. 3), the prior precision is proportional to the recurrent input strength (not weight) between two circuits (Fig. 13). When switching from stimulus OFF to ON, the firing rate of two circuits increases that implies the prior precision also increase gradually, and then the target distribution gradually shifts from the likelihood into the posterior with associative prior. Sorry that the page limit doesn’t allow us to incorporate such details.
> We’d like to clarify some potential confusion. Our usage of the “non-equilibrium” dynamics refers to the systems that are away from equilibrium, meaning they are not in a stable, time-independent steady state with detailed balance. And the non-equilibrium can exist in both autonomous and non-autonomous dynamical systems. Specifically, our non-equilibrium sampling refers to the transient phase of the circuit dynamics right after the stimulus is given (Fig. 2I) where the external feedforward inputs remain the same.
>
> ### The bivariate setup.
> Sorry that our figure may confuse you and we revise it in the revised manuscript. Fig. 3C emphasizes each 1D stimulus feature sample is individually generated from a network. Concatenating the samples from two networks and plotting it on a 2D plane (Fig. 3D, colored stochastic trajectory), the 2D samples converge to posteriors and their correlations contain the information of correlated prior.
> In Fig. A2D, we show the joint part of the precision of posterior sampled by network aligns with the ground truth while the mean of posterior also aligns with ground truth.
>
> ###  Bivariate Gaussian as a degenerate case
>
> You are correct. The bivariate Gaussian is degenerates since the determinant of matrix $L$ is 0. We will add a sentence in explaining that in the revised manuscript.

---

> ### Author Response · Authors · 2025-11-21
>
> ### Attractors
>
> Our attractor state (Eq. 2) refers to the attractor given non-zero feedforward inputs, which is a point attractor, precisely speaking, and is slightly different from the network’s self-sustained continuous attractors without feedforward inputs (like the persistent activity). In this way, the \bar $z_E$ in Eq. 2 (sorry, OpenReview doesn't support this math symbol) depends on the location of feedforward inputs, rather than a free parameter in continuous attractors.
> In our simulations, the recurrent weight $w_{EE}$  is usually smaller than $w_c$ and thus the network cannot hold persistent activity, i.e., no continuous attractors. However, even if $w_{EE}$ is larger than $w_c$ enabling the network to hold persistent activities, it doesn’t affect the posterior sampling in the circuit, in that its equilibrium sampling distribution is invariant with $U_E$ that increases with $w_{EE}$. This is numerically verified in Fig. 2C&E.

---

### Author Response · Authors · 2025-11-21
**Global reply**

## Significance and contribution of our study
Our theoretical neuroscience study aims to understand the algorithm of canonical neural circuits. An exact and analytical mapping between nonlinear recurrent circuit dynamics and Bayesian algorithms is largely missing due to complex, nonlinear circuit dynamics. Therefore, we perform comprehensive math analysis on an analytically solvable nonlinear circuit model, which for the first time builds __an analytical mapping between the nonlinear circuit and the NG sampling in the stimulus subspace__. These are the major theoretical contributions of the current study and are acknowledged by most reviewers.


## The novelty of the present paper compared with Sale 2024
The present study gains significant __algorithmic understanding__ as well as __technical/theoretical expansion__ compared with Sale 2024.
1.   Our work for the first time shows the canonical circuit with two types of interneurons implements __NG__ Langevin sampling (Sec. 4 and 5) and Hamiltonian sampling (Sec. 6) of the entire family of posteriors **without** any manual intervention of the circuit activities or parameters.
2. We further __analytically__ identify the __approximation strategy__ used in the circuit to estimate the Fisher information (FI), including the **regularization** in FI inversion (The recurrent E input in Sec. 4.2 and $\alpha$ in Eq. 5), and diagonal FI matrix approximation (Eq. 14). It is a surprise that the approximation strategy in neural circuit is the same as conventional ML algorithms.
3. We analytically identify an embedded the __annealing strategy__ in the circuit (Eq. 3b and Sec. 4.3).
- __Theoretical analysis__: This conceptual advance was only possible because we developed a more comprehensive analytical framework. Sale 2024 only projected the dynamics onto the position mode (Eq. 10 in Sale 2024), while the present study projected the circuit dynamics onto both position mode (Eq. 12a) and the __height__ mode (Eq. 12b). The height mode projection is the cornerstone for identifying the NG sampling in the circuit dynamics. We formally demonstrate that:
1. The height $U_E$ (Eq. 3b) acts as the **dynamic time constant** of the position dynamics (Eq. 3a).
2. It dynamically **encodes the Fisher Information** of the probability distribution.

## Our research route
Our research route is __from neural circuits to their internal models and algorithms__. We start from the circuit model widely used in neuroscience research and theoretically discover its internal model and algorithms. It is a complementary technical route compared with conventional approach starting by assuming (complex) generative models and seeking their circuit implementation, which, however, usually faces a dilemma that the derived circuit may lack biological plausibility. Since both research routes share a common goal of linking the circuit and the internal model and algorithms, and hence the __direct link__ in our study will be the one of the most valuable results to the field.
Even if our discovered Gaussian model is relatively simple, it presents a novel approach in the field and it can motivate the field to revise the circuit model to incorporate a more complex internal model which can be facilitated by our established analytical approach (see the Generalization section below). Consequently, the NG circuit sampling with an internal Gaussian model remains valuable to the field.

## Generalization: recurrent circuit as latent sampler in deep generative models
Our analytical mapping between circuit and algorithm provides a solid basis to generalize into more complex tasks. Due to the space limit, it is implausible to incorporate comprehensive theoretical analysis of NG sampling of non-Gaussian, multimodal distributions in the manuscript, so we add an Appendix G in the revised manuscript to discuss some mechanisms. In addition, to illustrate the generalization of our NG Gaussian posterior sampling circuit, we add a proof-of-concept example in a new section before the Discussion and a new Fig. 5 in the revised manuscript.
In this example, we propose the __NG Gaussian sampling circuit can be a latent sampler in deep generative models__. We are motivated by the convention in deep learning that the deep networks utilize encoders and decoders to map complex data distributions into simple Gaussian distributions in latent space. In this way, our NG Gaussian sampling circuit serves as a biologically plausible network implementation for a latent space sampler in deep generative models. Note that although deep generative models have achieved great success in applications, their whole network architecture still has a huge gap to neural circuits by lacking recurrent network architecture. Our analysis shows the NG sampling in the canonical circuit is maintained in the latent space in deep generative models (Fig. 5G).

---

### Meta-Review · Area_Chair_dbxn · 2025-12-11

**Summary:**

The paper tackles an ambitious and interesting question: whether canonical cortical circuits can be understood as implementing sophisticated sampling algorithms, and whether their intrinsic dynamics might naturally give rise to mechanisms resembling natural-gradient updates. Reviewers agreed that this questions is interesting and that the authors provide substantial analytical work in support of it. At the same time, they did not agree whether  the present manuscript convincingly establishes these claims.

A major unresolved issue is that the circuit implements only a diagonal approximation to the Fisher Information, which departs from true natural-gradient MCMC and produces clear performance gaps in multivariate settings. Reviewers remained unconvinced that this limitation can be dismissed, particularly given the strong claims in the title and abstract. There was also continued concern about the relationship between this submission and a closely related prior and concurrent manuscript built on the same circuit model which are sharing overlapping derivations and results.

Overall, I find the direction exciting and promising, but the current submission does not yet resolve the substantive concerns raised in the reviews. I do hope that a  revised and consolidated version can clarify these issues more decisively.

**Reviewer Concerns:**

see their reviews

**Reviewer Scores:**

I can not look into their heads, but attempted to make the best decision based on the information available.

---

### Decision · Program_Chairs · 2026-01-26

Reject